# Multi-layered dosage compensation of the avian Z chromosome by increased transcriptional burst frequency and elevated translational rates

Natali Papanicolaou [1,4], Antonio Lentini [1,4], Sebastian Wettersten[1], Michael Hagemann-Jensen [2], Annika Krüger [1], Jilin Zhang [1], Christos Coucoravas[1], Ioannis Petrosian[1], Xian Xin [1], Ilhan Ceyhan [1], Joanna Rorbach [1], Dominic Wright [3] & Björn Reinius [1] ✉

Sex-chromosome dosage poses a challenge for heterogametic species in maintaining the proper balance of gene products across chromosomes in each sex. While therian mammals (XX/XY system) achieve near-perfect balance of X-chromosome mRNAs through X-upregulation and X-inactivation, birds (ZW/ZZ system) have been found to lack efficient compensation at RNA level, challenging the necessity of resolving major gene-dosage asymmetries in avian cells. Through comprehensive allele-resolved multiome analyses, we examine dosage compensation in female (ZW), male (ZZ), and rare intersex (ZZW) chicken. Our data reveal that females upregulate their single Z chromosome through increased transcriptional burst frequency, mirroring mammalian X upregulation. Z-protein levels are further balanced in females through enhanced translation efficiency. Additionally, we present a global analysis of promoter elements regulating transcriptional burst kinetics in birds, revealing evolutionary conservation of the genomic encoding of burst kinetics between birds and mammals. Our study provides insights into the regulation of avian dosage compensation, and when considering all regulatory layers collectively, an unexpected similarity between avian and mammalian dosage compensation becomes apparent.

Vertebrate sex chromosome systems fall into two fundamental types, XX/XY and ZW/ZZ, defined by which sex is homogametic and which is heterogametic[1]. In most mammals, including mice and humans, females are homogametic and possess two large, gene-rich sex chromosomes (XX), while males are heterogametic, carrying one large and one degraded sex chromosome (XY). In ZW/ZZ systems, found in birds and reptiles, this relationship is inversed. Although the systems evolved independently, both originated from an autosomal chromosome pair and experienced degeneration of the non-recombining sex chromosome[2,3], which defines the heterogametic sex. This degeneration left the heterogametic sex (XY males and ZW females) with only a single copy of X/Z genes, resulting in an imbalance compared to the homogametic sex and the diploid autosomal gene expression network from which the sex chromosome originated. Pioneering theoretical

[1]Department of Medical Biochemistry and Biophysics, Karolinska Institutet, Stockholm, Sweden. [2]Department of Cell and Molecular Biology, Karolinska Institutet, Stockholm, Sweden. [3]AVIAN Behavioural Genomics and Physiology Group, IFM Biology, Linköping University, Linköping, Sweden. [4]These authors contributed equally: Natali Papanicolaou, Antonio Lentini. ✉e-mail: bjorn.reinius@ki.se

work by Susumu Ohno[4] proposed that cells must restore such an imbalance by a sex-chromosome-specific gene regulatory mechanism. Today, it is experimentally well-characterised that mammalian species such as human and mouse achieve dosage compensation at the transcriptional level by inactivating one X in females (X-inactivation) and upregulating the single active X chromosome (X-upregulation) in both sexes[5–7]. Conversely, the question of Z-chromosome dosage compensation remains debated to this day. Early studies in birds suggested little to no compensation, while most recent works reported Male:Female Z-RNA-expression ratios of 1.2-1.6, suggesting inefficient, gene-specific, compensation rather than a chromosome-wide effect, thus questioning the generality of Ohno's hypothesis[8–12]. Some delimited segments of Z behave differently, such as the male-hypermethylated (MHM) regions[13,14] which display female-specific expression. More recently, a strongly male-biased Z-linked microRNA, miR-2954, has been reported to target dosage-sensitive Z-linked genes[15–17], which contributes to dosage compensation. However, current data on Z-chromosome dosage compensation remain largely fragmented, often with partial biological information combined from different sources and studies, and no study to date provides allele-resolved single-cell data essential for understanding allelic expression dynamics central to this inquiry[18]. Finally, eukaryotic transcription occurs in stochastic bursts of RNA synthesis from each allele[19–23]. However, despite its crucial role in understanding expression regulation, the kinetics of transcriptional bursting and its genomic encoding remain entirely unexplored in birds, to our knowledge.

In this work, we apply a multimodal approach, combining chromatin analyses, transcriptional kinetics, ribosome profiling, and proteomics to uncover mechanisms of avian Z-chromosome dosage compensation, revealing Z-upregulation driven by increased transcriptional burst frequency and elevated translational rates.

## Results

### Z-chromosome upregulation throughout female tissues

To begin assessing Z-chromosome dosage compensation, we maintained Red Junglefowl (RJF; *Gallus gallus*) and White Leghorn (WL; *G. g. domesticus*) chicken breeds, as well as generated F1 hybrid offspring between the two, and performed bulk RNA-sequencing on female and male tissues (brain, liver, kidney, skin, ovary, and testis) (Fig. 1a, Supplementary Fig. 1a). We detected consistently imbalanced Z-chromosome RNA expression levels between males and females across tissues and independent of cross, with an average male-to-female Z-chromosome RNA level ratio of 1.57 across tissues (Fig. 1b, Supplementary Fig. 1b, and Supplementary Data 1). Similar male bias was observed across the Z-chromosome (Fig. 1c), with exception of the ~250kb-long MHM1 region[13], known to contain female-biased transcripts[8,24]. Z-linked genes retained from whole-genome duplication events in an ancient vertebrate ancestor (~450 MYA) are thought to be more dosage sensitive[25,26], and indeed, such genes showed lowered ratios in our RNA-seq data (-1.36, $p = 1.22 \times 10^{-7}$, Tukey-HSD ANOVA corrected for tissue type) whereas more recent human orthologs (-310 MYA)[27] or avian evolutionary strata (-100 MYA)[28] did not differ across tissues ($P_{human} = 0.96$, $P_{avian} > 0.11$) (Supplementary Fig. 1c). Male-to-female ratios were, overall, not associated with any apparent functional annotations in any of the somatic tissues (FDR > 0.05, GSEA Biological Processes).

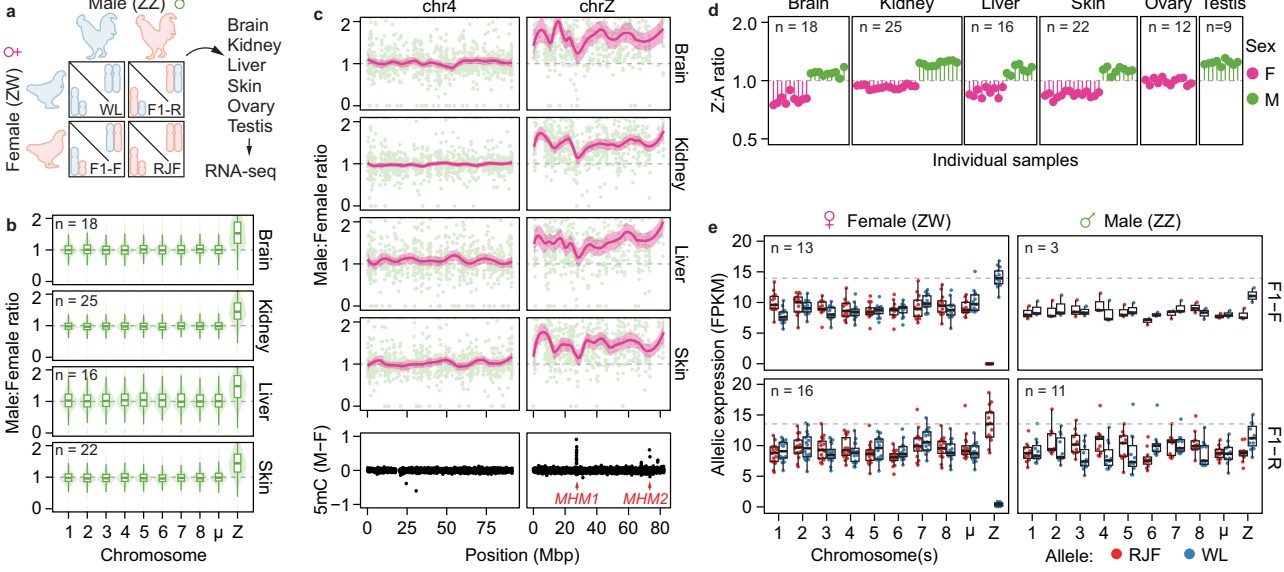

**Fig. 1 | Transcriptional upregulation of the female Z chromosome. a** Schematic representation of the experimental set up. RNA from brain, kidney, liver, skin, ovary and testis tissues was isolated from purebred WL (White Leghorn; blue) or RJF (Red Junglefowl; red) or F1 crossbred chickens (F1-Forward: RJF_mat x WL_pat or F1-Reverse: WL_mat x RJF_pat) and used for allele-resolved RNA-sequencing using bulk UMI Smartseq2. **b** Male:Female ratios of gene expression per tissue as boxplots over violin plots. Number of samples per tissue (n) shown in respective panels. Only expressed genes (average FPKM > 1; chr1 = 2055, chr2 = 1355, chr3 = 1219, chr4 = 1118, chr5 = 957, chr6 = 545, chr7 = 513, chr8 = 513, μ = 6045, chrZ=710 genes) were included in the analysis. μ denotes grouped micro-chromosomes 9-33. Data shown as median (center line), first and third quartiles (box limits) and 1.5x interquartile range (whiskers). **c** Male:Female gene expression ratios along chromosomes 4 and Z per tissue. Average ratio per gene shown in green, with a rolling average (LOESS) ± 95% confidence interval shown in pink. On the bottom panel,

Male:Female (M-F) 5mC (5-methylcytosine) enrichment based on bisulfite is shown for chromosomes 4 and Z, with male hypermethylated regions denoted in red. **d** Z:autosome ratios of bulk RNA-seq of WL, RJF and F1 chicken tissues. Each line and dot represents individual samples. Number samples (n) per tissue type shown in respective panels. Only expressed genes (FPKM > 1; chrA: n = 266-936, chrZ: n = 351) were included in the analysis. Female and male samples coloured in pink and green respectively. **e** Boxplots of allelic expression (FPKM) in female and male samples for each chromosome for F1 tissue samples (see **a**). RJF and WL alleles are shown in red and blue, respectively. μ denotes grouped micro-chromosomes 9-33. Number of samples (n) shown in respective panels. Only expressed genes (average FPKM > 1; 7047 (chr1 = 936, 2 = 628, 3 = 567, 4 = 554, 5 = 424, 6 = 301, 7 = 261, 8 = 266, 9-33 = 2758, Z = 351) were included in the analysis. Data shown as median (center line), first and third quartiles (box limits) and 1.5x interquartile range (whiskers).

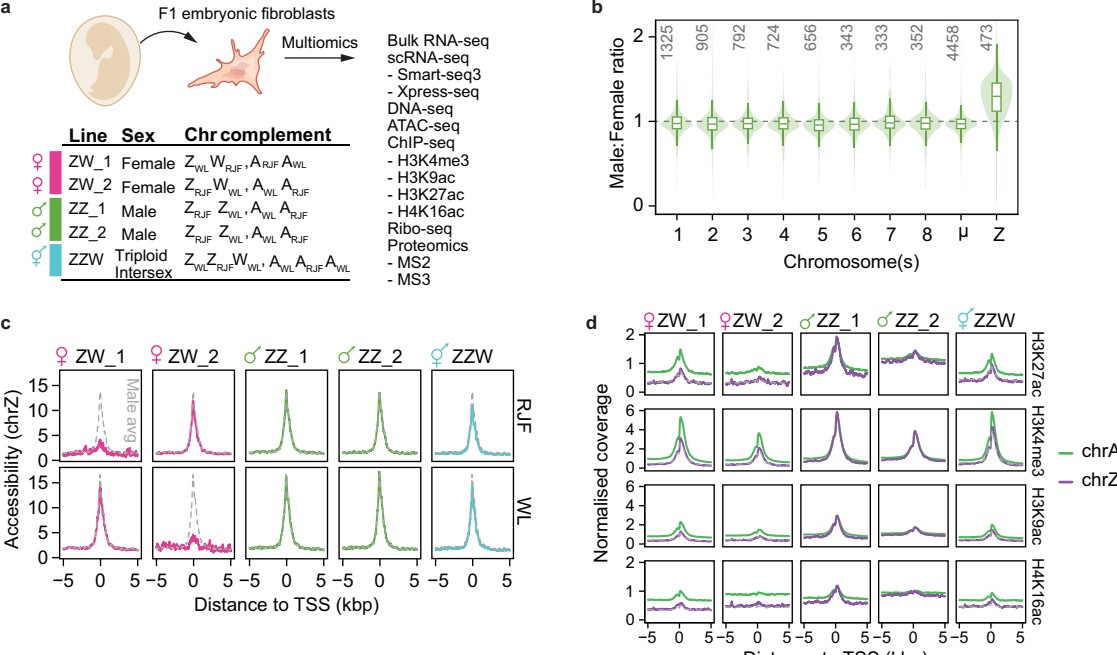

**Fig. 2 | Multiomics analysis of Z-chromosome upregulation. a** Schematic representation of the experimental set up. Primary chicken embryonic fibroblasts (CEF) were isolated F1 chicken embryos and used for multiomic characterisation. **b** Boxplots of Male:Female gene expression ratios per chromosome for chicken embryonic fibroblasts (CEFs), based on $n = 2$ male and $n = 2$ female samples (see **a**) from the average of 3 technical replicates. μ denotes grouped microchromosomes 9-33. Number of genes shown in respective panels. Only expressed genes (average FPKM > 1) were included in the analysis. Data shown as median (center line), first and third quartiles (box limits) and 1.5x interquartile range (whiskers). **c** Density plots of allele-resolved ATAC-seq signal enrichment at transcription start sites (TSS), per sample and allele, represented as means ± 95% CI obtained from 4 technical replicates for two female (ZW_1, ZW_2; pink), two male (ZZ_1, ZZ_2; green), and one triploid (ZZW; teal) sample. **d** Density plots of quantitative ChIP-seq signal enrichment around transcription start sites (TSS), per histone modification and sample, represented as means ± 95% CI obtained, based on 3 technical replicates. Autosomal signal shown in green and Z-chr signal in purple.

## Decoding avian dosage compensation allele by allele

Importantly, the observed male-to-female expression ratios (~1.57) deviate from the 2-fold ratio expected in complete absence of Z-dosage compensation between ZZ (male) and ZW (female) genotypes. In line with this, comparing Z-linked expression to diploid autosomes (AA) per sample showed that female Z expression (Z:AA) was higher than expected for all tissues (Fig. 1d). To explore Z-dosage compensation at the allele-specific level, we called genetic variants from the pure WL and RJF parental breeds, allowing allelic expression analyses in F1 offspring. After filtering and using N-masking to minimise mapping bias (see **Methods**), we retained 193,130 allele-informative variants covering 84% of expressed genes across all chromosomes, enabling high-resolution allelic inference (Supplementary Fig. 1d-e). Whereas there was a slight-bias towards the WL Z chromosome in F1 males compared to autosomes, we did not observe an overrepresentation of differentially expressed Z-linked genes in pure WL breeds compared to pure RJF (OR = 1.05, $p = 0.72$, Fisher's exact test, Supplementary Data 2). Interestingly, utilising our allelic expression measurements, we observed that the single Z chromosome in female tissues was distinctly upregulated compared to individual autosomal alleles as well as compared to the separate transcriptional output of each male Z allele (Fig. 1e, Supplementary Fig. 1f), indicating that partial dosage compensation is achieved through hyperactivation of the female Z chromosome. Intriguingly, this observed female-specific Z-chromosome upregulation resembles X-upregulation seen in both male and female mammals[5,29–31].

## Chromatin state of Z-chromosome dosage compensation

To extensively characterise Z-upregulation, we derived primary F1 chicken embryonic fibroblast (CEF) cell lines from eggs of RJF/WL intercrosses and performed a comprehensive array of omics profiling

(Fig. 2a). During analysis, we noticed that one CEF cell line expressed chromosome W in addition to two Z alleles, which we resolved as triploid intersex ($Z_{WL}Z_{RJF}W_{WL}$: $A_{WL}A_{RJF}A_{WL}$; 3n) by DNA-seq and karyotyping (Supplementary Fig. 2a-e, **Source data**). Triploidy is a naturally viable, but exceedingly rare, genotype in chickens (0.1–0.5%) arising from chromosomal nondisjunction during oogenesis[32] (Supplementary Fig. 2f), and, curiously, was first described by none other than Susumu Ohno[33]. It has been previously suggested that the presence of a W chromosome may contribute to the control of Z chromosome upregulation[34], and ZZW individuals have been previously shown to express the female-specific Z-linked MHM region[13]. The presence of triploid cells thus enabled us to assess the extent of Z-linked dosage compensation across diploid males (ZZ: AA), diploid females (ZW: AA), and triploid intersex individuals (ZZW: AAA), as well as to uncouple a) the effect of the W chromosome on Z-upregulation, in the presence of two copies of the Z chromosome, and b) the effect of biological sex on Z-upregulation. Expression relative to autosomes revealed that Z-linked expression was partially compensated in both ZW and ZZW genotypes (Supplementary Fig. 3a). Interestingly, whereas RNA-seq of CEF cell lines reconfirmed Z-upregulation in ZW females on par with in vivo tissues (mean fold-change 1.54 in CEFs, Fig. 2b), no such upregulation was observed in ZZW, but instead, a marked buffering of autosomal gene expression (Supplementary Fig. 3b), i.e. less than linearly additive, as has been observed for other species[35], suggesting that Z-chromosome upregulation is not dependent on the presence of a W chromosome, as has been previously hypothesised[34]. On the transcriptional level, intersex CEFs showed similar gene expression patterns as females, with high expression of W-linked *HINTW* and *SPINW*, believed to be involved in ovarian development, and female-biased Z-linked *BHMT2* expression (Supplementary Data 3) in agreement with observations

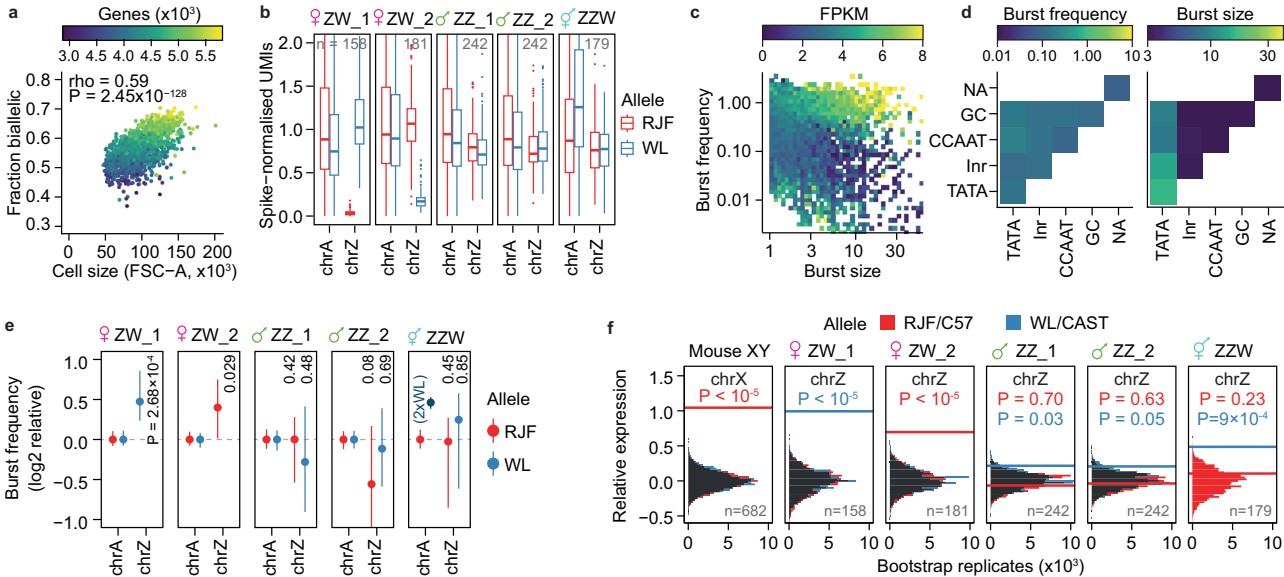

**Fig. 3 | Avian transcriptional bursting and Z chromosome upregulation.**
**a** Scatterplot of number of biallelically expressed genes (average FPKM > 20, 1033-3579 genes) per cell (n = 1382), by cell size (FSC-A) based on FACS and single-cell RNA-seq (Smart-seq3) data. Point colours are denoted as genes with FPKM > 1 per cell. **b** Boxplots of allele-resolved spike-in normalised UMI counts of single-cell RNA-seq (Xpress-seq) of CEFs per sample and allele. Number of single cells (n) shown in the respective panels. Only expressed genes (average FPKM > 1, chrA = 8467, chrZ = 424 genes) were included in the analysis. Data shown as median (center line), first and third quartiles (box limits), and 1.5x interquartile range (whiskers). **c** Scatter plot of burst frequency (y-axis) and burst size (x-axis) inferred for expressed genes (7392 genes, average FPKM > 1) in CEFs, coloured based on FPKM-normalised expression level. **d** Heatmaps of core promoter elements and their correlation with burst

frequency (left) or burst size (right) for expressed genes (7392 genes, average FPKM > 1). **e** Allele-resolved, log2-relative burst frequency per cell for autosomal (chrA = 4586) and Z-linked (chrZ = 229) genes, plotted as median ± 95% CI. RJF and WL alleles are shown in red and blue, respectively. P-values from two-tailed Mann-Whitney U tests are shown. **f** Relative expression levels of autosomal and X-linked genes in mouse and Z-linked genes in chicken. Histogram of medians of relative expression in FPKM of randomly subsampled genes, compared to the median of chrX for mouse (chrA genes = 10025 genes, chrX = 276 genes) or chrZ for chicken (chrA = 7180 genes, chrZ = 332 genes), based on scRNA-seq in mouse fibroblasts[50] and CEFs, with the number of cells used shown in black. P-values denote a one-tailed bootstrap test with 1×10⁵ permutations (no adjustments for multiple comparisons). C57 (C57BL6/J) and CAST (CAST/EiJ) denotes mouse alleles.

that ZZW intersex chicks phenotypically resemble females until week 20 post hatching[36].

To investigate whether the chromatin landscape of the Z chromosome contributes to Z-upregulation, we began by assessing chromatin accessibility in CEF lines using ATAC-seq. Similar to our previous findings of X-upregulation in mice[31], chromatin accessibility was not increased on the upregulated Z allele despite pronounced Z-upregulation on the transcriptional level (Fig. 2c and Supplementary Fig. 4a–c), suggesting limited involvement of chromatin accessibility in regulating Z-upregulation. Through a global analysis of transcription factor footprinting using ATAC-seq data, we examined differential transcription factor binding by performing pairwise comparisons of the Z chromosome and autosomes between the sexes. We observed preferential enrichment of the FOX transcription factor family on the Z chromosome and autosomes in males, while the GATA transcription factor family was enriched on the Z chromosome and autosomes in females (Supplementary Fig 4d, e, Supplementary Data 4). E-box-binding transcription factors were enriched on female and intersex Z chromosomes compared to the male Z, whereas this difference was not observed for autosomes (Supplementary Fig. 4d, e, Supplementary Data 4). Notably, E-box motif sequences were not enriched on the Z chromosome compared to autosomes (Supplementary Data 5) potentially, hinting to a sex-specific role in Z-gene regulation, rather than the female-specific Z-upregulation we observed, something that could be further explored in future studies. We next performed multiplexed quantitative ChIP-seq[37] (EpiFinder) for four permissive histone modifications (H3K4me3, H3K9ac, H3K27ac and H4K16ac) that correlated with promoter and enhancer features (Supplementary Fig. 5a-b). Although these modifications were overall associated with gene expression

levels (Supplementary Fig. 5c), they were not associated with Z-upregulation (Fig. 2d).

The intersex CEF cell line and allelic resolution allowed us to explore the MHM region from a new angle. As expected, we detected MHM-expression in ZW females and the lack thereof in ZZ males. Intriguingly, intersex ZZW displayed accessible chromatin and biallelic RNA expression (Supplementary Fig. 6). This confirmed previous hybridisation experiments[13], indicating that MHM expression is enabled by the presence of the W rather than controlled by Z-chromosome dosage.

### Dynamic random monoallelic expression in birds

To explore Z-upregulation at cellular allelic regulation, we performed full-transcript-length scRNA-seq using Smart-seq3[38] deep-sequencing of CEF cell lines (Supplementary Fig. 7a). These data reconfirmed female-specific Z-chromosome upregulation, but here, importantly, at the level of individual allele in single cells (Supplementary Fig. 7b). Notably, biallelic expression of the MHM region was observed in intersex ZZW cells (Supplementary Fig. 6c), thus demonstrating its expression from the maternal and paternal alleles. Single-cell resolution also allowed us to explore how general cell-intrinsic features associate with RNA expression output in birds, where cell size was highly correlated with genes detected (Spearman's rho = 0.6, $p = 3.47 \times 10^{-133}$) and number of RNA molecules per cell (Spearman's Rho = 0.5, $p = 3.03 \times 10^{-66}$) (Supplementary Fig. 7c). Furthermore, cell size was also associated with the fraction of genes expressed biallelically at the snapshot of sampling due to stochastic allelic transcription (Spearman's rho = 0.59, $p = 1.05 \times 10^{-132}$) (Fig. 3a). In summary, our data showed that the same principal laws of cell scaling and dynamic random monoallelic expression due to transcriptional bursting apply in birds as in mouse and human[39–41].

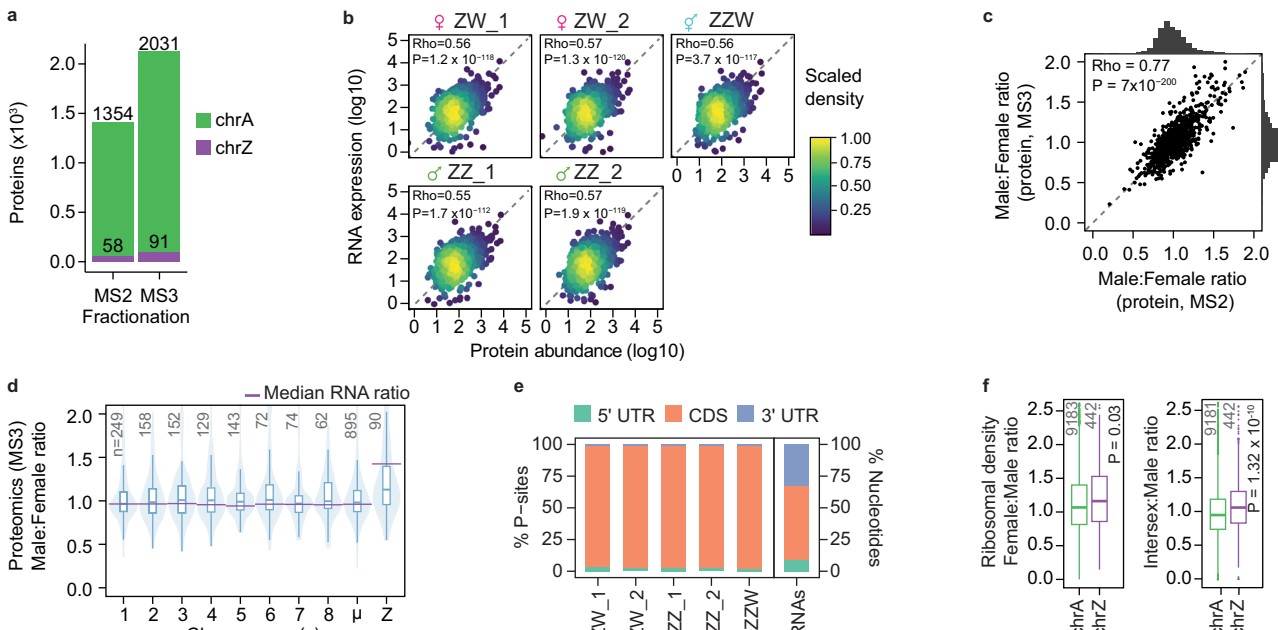

**Fig. 4 | Post-transcriptional Z-upregulation through elevated translation rate.**
**a** Bar plot showing the number of unique autosomal (chrA; green) and Z-linked (chrZ; purple) proteins identified using tandem mass spectrometry (MS2) or triple-stage mass spectrometry (MS3) in chicken embryonic fibroblast (CEF) lines ($n = 2$ female, $n = 2$ male and $n = 1$ triploid intersex samples, average of 3 technical replicates). **b** Scatterplots of RNA expression (y-axis, bulk RNA-seq) and protein abundance (x-axis; MS2) in CEFs, per sample, with coloured by density. Spearman's Rho and p-value based on a two-tailed Spearman rank correlation test shown for each sample. **c** Scatterplot of Male:Female ratios of protein abundances based on MS2 (x-axis) and MS3 (y-axis) proteomic measurements for proteins detected in both MS2 and MS3 datasets ($n = 1002$), with histograms of distributions shown for MS3 (y-axis) and MS2 (x-axis). Spearman's Rho and p-value based on a two-tailed Spearman rank correlation test shown over the scatter. **d** Boxplot over violin plots of Male:Female ratios, of protein abundances (MS3) per chromosome in CEFs. The number of genes included in the analysis per chromosome is shown in grey. Purple

vertical lines indicate median Male:Female ratios of gene expression based on bulk RNA-seq in CEFs. Data shown as median (center line), first and third quartiles (box limits) and 1.5x interquartile range (whiskers). μ denotes grouped micro-chromosomes 9-33. Data collected from 3 technical replicates. Number of samples (n) defined as: CEF lines derived from $n = 2$ female and $n = 2$ male embryos. **e** Bar plots displaying the percentage of ribosomal P-sites in 5' untranslated region (UTR), coding sequence (CDS) and 3' UTR of mRNA transcripts per sample, with the expected read distribution based on random fragmentation displayed on the right. **f** Female:Male and Intersex:Male ratios of translation efficiency, calculated as gene expression-normalised ribosome-protected footprint (RPF) FPKM counts for autosomes and chrZ. Based only on expressed genes (RPF > 1 & RNA FPKM > 1; number of genes shown in panel). Number of samples (n) defined as: CEF lines derived from $n = 2$ female, $n = 2$ male and $n = 1$ triploid embryos. P-values based on two-tailed Mann-Whitney-U test. Data shown as median, first and third quartiles and 1.5x interquartile range (IQR).

## Uncovering avian burst kinetics

To enable direct comparison between diploid and triploid expression levels we performed Xpress-seq on all five CEF cell lines. This method, derived from Smart-seq3xpress[42], enables full-length scRNA-seq and incorporates UMI-containing exogenous spike-in RNA[43], allowing for precise counting of original mRNA molecules in single cells (Supplementary Fig. 8). This re-confirmed female-specific Z-upregulation and, importantly, showed that ZZW lacked transcriptional Z-upregulation (Fig. 3b). At the single-cell level, eukaryotic transcription is inherently stochastic and occurs in short bursts of activity from the individual alleles[41,44]. Transcriptional kinetics can be represented by burst frequency (rate of transcription pulses) and burst size (average number of molecules produced per burst) within the two-state telegraphic model of transcription. We and colleagues, previously characterized the genomic encoding of transcriptional burst kinetics in mouse from allelic scRNA-seq data[19]. Moreover, we showed that X-upregulation in mouse is primarily driven by increased transcriptional burst frequency, and not increased burst size[30,31]. Here, we sought to characterise transcriptional kinetics in birds for the first time. To this end, we inferred parameters of transcriptional bursting (burst frequency: $k_{on}$ and burst size: $k_{syn}/k_{off}$) for each allele using our Xpress-seq data (see **Methods** and Supplementary Fig. 9a, b). First, we established the principles of burst kinetics in chicken which mirrored what is observed in humans and mice[19,45] (Fig. 3c). Specifically, we found burst size to be driven by the presence of TATA-box in the core promoter, with

Initiator elements (Inr) having a small additive effect (Fig. 3d), consistent with the notion that TATA promoters show high rates of continuous transcription[46,47]. In contrast, burst frequency was associated with cis-regulatory activity through permissive histone modifications (Supplementary Fig. 9c, d), as has previously been shown in mice[48]. Importantly, our data suggest that the control principles of burst kinetics are universally shared across mammals and birds, with the genomic encoding of transcriptional bursting being highly conserved among vertebrate species. We next explored the kinetic modulus of Z-upregulation, observing that burst frequency was increased on Z relative to autosomes in female ZW cells carrying either Z allele ($P_{WL-allele} = 2.68 \times 10^{-4}$, $P_{RJF-allele} = 0.029$, Mann-Whitney-U test) whereas burst frequency was not increased in Z alleles in male ZZ and intersex ZZW cells (RJF alleles analysed in triploid cells) (Fig. 3e). Conversely, burst size remained close to autosomal levels for all cells ($p > 0.41$) (Supplementary Fig. 10a), indicating that transcriptional Z-upregulation is primarily driven by burst frequency. It should be noted that inference of transcriptional kinetics is only robust for individual alleles[19], necessitating allelic scRNA-seq data herein provided, and allowing accurate kinetic inference of RJF alleles but not WL alleles in our triploid CEF cell line.

Given the similarities in transcriptional mode of Z-upregulation in chicken to X-upregulation in mice, we sought to compare the two directly. To compensate for gene content differences between chromosomes and species, we utilised a bootstrapping approach[31] to

compare gene-level transcriptional metrics. The degree of upregulation was nearly identical in chicken (ZW) and mouse (XY) fibroblasts (Fig. 3f, Supplementary Fig. 10b), suggesting similar dynamics of transcriptional upregulation between evolutionarily unrelated sex chromosomes.

### A second layer of dosage compensation: Post-transcriptional regulation

Despite the apparent tolerance to large Z-linked RNA expression differences between sexes, sex chromosome aneuploidies are embryonic lethal in chicken[34,49], begging the question whether the -1.57 male-to-female RNA expression imbalance might resolve at the proteomic level. To investigate this, we performed tandem mass spectrometry for the CEF cell lines using both MS2 and MS3 spectra (see **Methods**). Through additional fractionation (MS3), we were able to improve detection sensitivity and identified up to 2122 proteins per line including up to 91 Z-encoded proteins (Fig. 4a). Protein abundances showed good agreement with RNA-seq expression and between MS2 and MS3 fractionations (Spearman's Rho = 0.56, $p = 3.4 \times 10^{-102}$, Fig. 4b, Spearman's Rho = 0.77, $p = 7 \times 10^{-200}$, Fig. 4c), Interestingly, Male:Female protein abundance ratios were significantly diminished for the Z chromosome compared to RNA-level ratios ($p = 1.45 \times 10^{-5}$, one-sample Wilcoxon test), reaching only a 1.2-fold difference (Fig. 4d and Supplementary Fig. 11a). Indeed, gene-wise comparisons suggested that female (ZW) samples produce more Z-encoded protein per mRNA expression unit (Supplementary Fig. 11b). The intersex (ZZW) sample had higher Z-protein levels compared to females but also displayed diminished differences at the protein level (Supplementary Fig. 11c).

### Increased ribosome occupancy on Z-linked transcripts

Seeking to explore whether the effect observed on the protein level could be explained by differences in translation efficiency, we performed ribosome profiling (Ribo-seq) on the five CEF cell lines. Using ribosome-protected fragment counts normalised to RNA expression for each gene (see **Methods** and Fig. 4e, Supplementary Fig. 12a, b), we calculated translation efficiency (TE) rates for autosomes and the Z-chromosome in the three sexes. Indeed, female and intersex samples displayed higher TE rates for the Z chromosome than for autosomes compared to males ($P_{ZW} = 1.11 \times 10^{-4}$, $P_{ZZW} = 8.27 \times 10^{-5}$, $P_{ZZ} = 0.14$, Mann-Whitney-U test) (Supplementary Fig. 12c), suggesting increased ribosome occupancy on female and intersex Z-linked transcripts. To compare ribosome occupancies in a gene-wise manner, we calculated Female:Male and Intersex:Male ratios for expressed genes and across different expression cutoffs, which further confirmed significantly higher TE for chromosome Z in female and intersex chicken ($P_{F:M} = 0.03$, $P_{I:M} = 1.32 \times 10^{-10}$, Mann-Whitney-U test) (Fig. 4f). Although the detected effect was modest, it is interesting that increased Z translation in birds parallels previous reports of an increased translational efficiency of X-linked transcripts in eutherian mammals[50,51], again highlighting shared molecular solutions in Z- and X-upregulation. Of particular note, while female-to-male RNA levels are near-perfectly balanced in mouse and human, X:AA ratios are not[30,31], explaining the need of translational upregulation in mammals carrying one active X chromosome per cell (i.e. both females [XaXi] and males [XaY]) similar to female (ZW), but not male (ZZ), birds.

### A partial role of miR-2954 in dosage compensation

Our findings that Z-upregulation is driven by increased burst frequency and translation rate do not exclude the involvement of additional mechanisms at play in avian dosage compensation. The male-biased miR-2954 has been proposed to target a subset of dosage-sensitive Z-linked mRNAs[15–17], and thus we explored to what degree miR-2954 might have contributed to dosage compensation in our data. Quantification of miR-2954 by RT-qPCR showed considerably higher expression in male CEF cell lines and in vivo tissues compared to

females, while intermediate miR-2954 levels were observed in the ZZW genotype (Supplementary Fig. 13a, b, **Source Data**). It is worth noting that while miR-2954 expression was male-biased in all tissues examined, miR-2954 was detectable and modestly expressed in all female tissues, with male-to-female fold-changes ranging 4-14-fold, highlighting significant tissue expression variability (Supplementary Fig. 13a, b, **Source Data**). Comparing expression ratios in RNA-seq for genes containing or lacking miR-2954 target sequence (29% CEF-expressed Z genes contained the target sequence; **Methods**), we indeed found lowered Male:Female ratios for miR-2954 targets on Z but not autosomes at the RNA level, while intersex comparisons suggest that ZZW behave more like males than females in this aspect (Supplementary Fig. 13c, d, **Source Data**). Yet, complete dosage compensation of Z-linked miR-2954 targets was not observed in either male or intersex samples. Thus, while miR-2954 may provide partial relief by reducing RNA levels of select Z transcripts in males, we conclude that chicken rely on a multi-layered dosage compensation strategy.

## Discussion

In this study, we aimed to uncover the mode and degree of dosage compensation in birds. Using bulk RNA-seq on pure-line and F1 allele-resolved chicken tissues, we demonstrated that sex chromosome dosage compensation is achieved through chromosome-wide transcriptional upregulation of the single Z allele in females, achieving 40-50% dosage compensation at the RNA level. We investigated transcriptional burst kinetics using high-sensitivity, allele-resolved scRNA-seq on primary chicken fibroblasts, establishing that Z-upregulation is driven by increased burst frequency, but not size, resembling the kinetic mode of the evolutionarily distinct X-upregulation observed in mouse[30,31]. For the first time, we characterized the genomic encoding of burst kinetics in birds and found that, despite -310 million years of evolutionary separation from mammals, core promoter elements controlling transcriptional burst size remain conserved. Specifically, TATA and Inr promoter elements correlate with increased burst size, suggesting their control over transcriptional kinetics are fundamental across vertebrates.

Chromatin accessibility and permissive histone modifications have been suggested to at least partially underlie X-upregulation in mouse[52–54]. However, despite a pronounced compensation on the transcriptional level, using allele-resolved ATAC-seq and quantitative ChIP-seq of selected markers, we observed no differences in accessibility or chromatin state between female and male Z-chromosomes. This is in line with previous reports of non-linear relationships between chromatin state and gene expression in sex-chromosome upregulation[31,52,53]. The absence of chromatin accessibility differences between an upregulated and non-upregulated allele of both the mammalian X and avian Z chromosomes highlights another layer of similarity in their dosage compensation mechanisms.

The serendipitous inclusion of a natural, but rare, triploid intersex chicken (ZZW) enabled us to uncouple the effect that the W versus having two Z chromosomes may have on Z-upregulation. Notably, while both female and intersex samples carry a W chromosome, we found no evidence of transcriptional Z-upregulation in the ZZW genotype. Conversely, we observed that intersex cells displayed a buffering of autosomal gene expression, which would improve the overall stoichiometric balance, rendering Z-upregulation redundant, across both autosomal and sex chromosomes, and therefore contribute to the viability of these individuals. Thus, we conclude that Z-upregulation by increased transcriptional burst frequency in female cells results from carrying only one Z allele, and not from the presence of a W chromosome, contrasting to the proposition of a locus on the W chromosome being the main driver of Z-upregulation[34]. Despite the presence of two Z chromosomes in intersex, we found the MHM region to be accessible and transcriptionally active in both females and

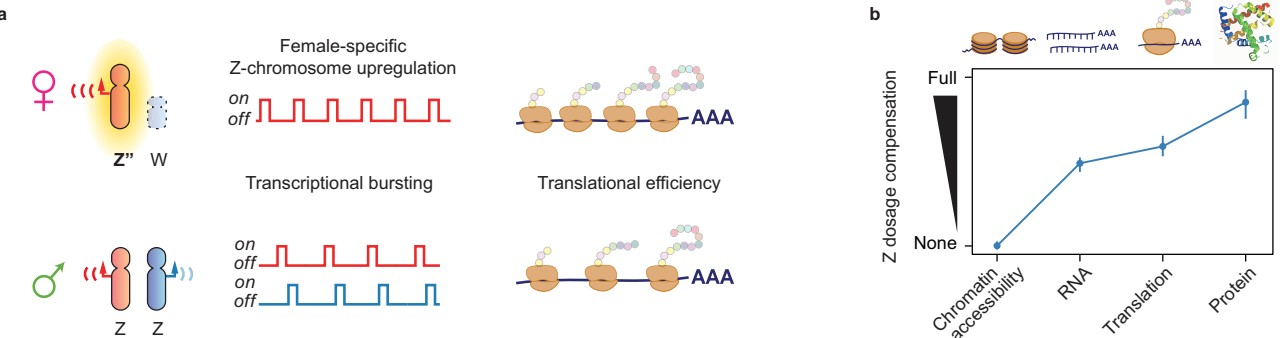

**Fig. 5 | Multi-layered dosage compensation of the avian Z-chromosome.**
**a** Transcriptional upregulation of the single female Z chromosome is driven by increased transcriptional burst frequency resulting in a transcriptionally hyperactive female Z chromosome (Z"). In addition, dosage compensation is further achieved through increased translational efficiency of Z-linked transcripts in females, mediated by increased ribosomal occupancy compared to Z-linked transcripts in males. **b** Schematic representation of a multi-layered model of avian Z-chromosome dosage compensation. At the genomic level, the single female Z chromosome does not display enhanced accessibility. At the gene expression level, compensatory transcriptional upregulation of the female Z is apparent, and driven by increased transcription burst frequency. Z-linked transcripts display higher translational efficiency, partially contributing to an overall rebalancing and near-complete dosage compensation between ZZ males and ZW females at the proteomic level. Plotted as median ± 95% confidence interval (C.I) of Male:Female gene expression ratios for each modality based on ATAC-seq (chrZ genes = 548), bulk RNA-seq (chrZ genes = 501), Ribo-seq (chrZ genes = 443) and proteomics (chrZ genes = 90) data. Male:Female ratios based on chicken embryonic fibroblasts derived from $n = 2$ male and $n = 2$ female F1 chicken embryos.

intersex, indicating that MHM expression is dependent on the presence of a W and not the number of Z chromosomes.

Despite transcriptional dosage compensation, protein stoichiometries are ultimately crucial for cellular and organismal fitness and survival. We found that the upregulation of the female Z chromosome is further enhanced at the proteome level, achieving a significant dosage rebalancing (~80%) between males and females. Our proteomic experiment was similar to that of a recent study reporting the complete lack of sex differences in Z-protein levels in chicken and platypus[55]. However, in contrast, our mass spectrometry data revealed a small but significant male bias remaining at the protein level (male-to-female expression ratio: 1.2-fold, $p = 8.6 \times 10^{-7}$, Mann-Whitney-U test). This discrepancy in results may stem from differences in tissues analysed (liver and heart versus fibroblasts) or the increased statistical power of our experiments, capturing more than twice the number of proteins and maintaining a comparatively strong correlation between RNA and protein ratios (Spearman's rho = 0.41, $p = 1.92 \times 10^{-82}$, MS3 data) (Supplementary Fig. 11b and Supplementary Fig. 14a, b).

What mechanisms drive additional dosage compensation at the proteome level? Our Ribo-seq data indicates that this is least partially achieved by higher ribosomal density, and thus increased translational efficiency, of Z-transcripts in females. Intriguingly, similarly Z-enriched ribosomal density was observed in ZZW intersex cells suggesting the involvement of W-linked factors. This could include mechanisms involving lncRNAs acting as sponges to sequester miRNAs[56] targeting Z-linked transcripts, thereby preventing mRNA-miRNA interactions that result in translation inhibition, or even miRNAs that enhance translation of Z-linked transcripts depending on their localisation and 5' UTR structures[57–59]. Increased translation efficiency of Z in female, but not male, cells contrasts to the situation in mammals, where X translation is elevated in both sexes[50,51]. Notably, however, while X-linked RNA levels are almost fully balanced between mammalian females and males, the transcriptional upregulation of their single active X ($X_aX_i$ and $X_aY$, respectively) does not fully reach the biallelic RNA levels of autosomes[30,31]. Consequently, the enhanced Z translation rate alongside transcriptional Z upregulation in female birds (ZW) parallels the scenario in mammalian cells, across the sexes. In contrast, male birds (ZZ) exhibit balanced Z-to-autosome RNA levels, deviating from this pattern. Thus, our data highlight fundamental principles of dosage compensation that apply across evolutionarily distinct sex-chromosome systems.

In conclusion, our study reveals a complex interplay between transcriptional and translational mechanisms that work synergistically to achieve extensive dosage compensation of the Z chromosome (Fig. 5), and when considering all regulatory layers collectively, an unexpected similarity to mammalian dosage compensation emerges.

## Methods
### Ethics statement
All animal experimental procedures were performed in accordance with Karolinska Institutet's and Linköpings Universitet's guidelines and approved by the Swedish Board of Agriculture (permit 16110-2020 Jordbruksverket).

### Tissue RNA isolation
Brain, kidney, liver, skin, ovaries and testes were isolated from adult chickens. RNA isolation was performed using the phenol-chloroform extraction method. Briefly, 50–100 mg of tissue was isolated and homogenised using 1 ml of TRIzol reagent [Thermofisher, cat. number: 15596018] and a tissue homogeniser. To precipitate the RNA, 500 µl of isopropanol was added per 1 ml of TRIzol and the mixture was incubated for 10 min followed by centrifugation at 12000 x $g$ at 4 °C for 10 min. To wash the RNA, the pellet was resuspended in 1 ml of 75% ethanol per 1 ml of TRIzol used and centrifuged for 5 min at 7500 x $g$ and 4 °C. The supernatant was discarded, and the pellet was left to air-dry for 5–10 min. The pellet was resuspended in 50 µl of nuclease-free water, incubated at 55 °C for 15 min and stored in −80 °C. The concentration was measured using a Nanodrop 2000 instrument.

### Derivation and culturing of cell lines
Chicken embryonic fibroblasts (CEFs) were derived from 10- to 13-day-old chicken embryos. Briefly, eggshells were swabbed with 70% ethanol before removing the round end of the egg. Using sterile forceps, the chorioallantoic membrane was snipped and the chicken embryos were removed. The embryos were sacrificed by decapitation and were subsequently finely minced with sterile scissors. The tissue was further dissociated by incubation in 0.25% trypsin at 37 °C for 15 min. The supernatant was removed and centrifuged at 300 x $g$ for 5 min and the cell pellet was washed twice in 1x PBS [Gibco, cat. number: 10010023]. The pellet was resuspended in 10 ml of complete media (10% Foetal Bovine Serum [Gibco, cat.number: A5256701], 100U/ml Penicillin – 100 µg/ml Streptomycin [Gibco, cat. number: 15140148], 1 mM non-

essential amino acids [Gibco, cat. number: 11140050], 1 mM Sodium pyruvate [Gibco, cat. number: 11360070]) and plated in Petri dishes pre-coated with 0.1% gelatin solution. CEFs were incubated at 37 °C in a humidified 5% CO2 incubator until confluency was reached. Upon confluency, cells were passaged using the TrypLE dissociation reagent [Gibco, cat. number: 12604021] and re-plated in 0.1% gelatin-coated plates.

## Quantitative Real-Time PCR - miR-2954 (CEF and tissues)

**RNA isolation.** RNA isolation was performed using the phenol-chloroform RNA extraction method. Briefly, 350 µl of TRIzol [Thermofisher, cat. number: 15596018] reagent was added to pellets of chicken embryonic fibroblasts grown to 80% confluency, followed by addition of 70 µl of chloroform to allow for phase separation. After a room temperature incubation of 5 min, the samples were centrifuged for 15 min at 12000 x $g$ and 4 °C. The aqueous phase was carefully isolated to avoid any contact with the DNA- and protein-containing inter- and organic phases and transferred to a new microcentrifuge tube. To precipitate the RNA, 175 µl of isopropanol was added and the mixture was incubated for 10 min at RT, followed by centrifugation at 12000 x $g$ and 4 °C for 10 min. The supernatant was discarded, and the RNA pellet was washed with 350 µl of 75% EtOH, followed by centrifugation at 7500 x $g$ and 4 °C for 5 min. Finally, the pellets were resuspended in 40 µl of RNase-free water and concentration measured on a Nanodrop 2000 instrument.

**Reverse Transcription.** Reverse transcription was performed using the PrimeScript RT reagent kit with gDNA eraser [Takara, cat. number: RR047A] according to the manufacturer's instructions and as previously described in Cheng et al. 2022. Briefly, 1 µg of RNA was treated with 1 µl of gDNA eraser and incubated for 2 min at 42 °C, followed by reverse transcription using either Primescript's RT primer mix at 37 °C for 15 min, 85 °C for 5 s and 4 °C hold or a miR-2954-specific stem-loop primer (RT-miR-2954-stem-loop primer: 5'- CTCAACTGGTGTCGTG-GAGTCGGCAATTCAG TTGAGTGCTAGGA-3') at 42 °C for 15 min, 85 °C for 5 s and 4 °C hold.

**qPCR.** qPCR was performed using the PowerUp SYBR green mastermix reagent [Applied Biosystems, cat. number: A25741] according to the manufacturer's instructions. Specifically, 1 µl of 1:10-diluted cDNA was used as input in a 10 µl reaction including 5 µl SYBR green reagent, 3 µl RNase-free water and 0.5 µl forward and reverse primers at 10 µM. The reaction was performed on an Applied Biosystems Real-Time PCR instrument using the following program: 50 °C for 2 min, 95 °C for 2 min, followed by 40 cycles of: 95 °C for 3 s and 60 °C for 30 s. Ct values from RT-qPCR experiments are available in the Source Data.

## Bulk RNA-sequencing using UMI-Smart-seq2

### Library preparation

**Cell lysis.** Bulk Smart-seq2 was performed as previously described with slight modifications[30]. Specifically, 2 ng of purified tissue RNA was added to 3 µl of Smart-seq2 lysis buffer (1 µM oligo-dT primer [5'-Biosg//idSp//idSp//idSp/ACGAGCATCAGCAGCATACGAT30VN; IDT], 0.5 mM (each) dNTPs [Thermo Scientific, cat. number: R0193], 0.2% Triton-X-100 [Sigma Aldrich, cat. number: T8787], 1 U/µl RNase inhibitor [Takara, cat. number: 2313 A]. To ensure RNA denaturation, the samples were incubated at 72 °C for 3 min and immediately placed on ice.

**cDNA synthesis.** 5.7 µl of reverse transcription mastermix (1x Superscript II first-strand buffer [Thermo Fisher Scientific, Invitrogen, cat. number: 18064071], 5 mM betaine [Sigma Aldrich, cat. number: B0300], 6 mM MgCl2 [Thermo Fisher Scientific, Invitrogen, cat. number: AM9530G, 1 µM TSO [5'-biotin-AGAGACAGATTGCGCAATGH HHHHHrG+GG-3'; IDT], 1.7U/µl of recombinant RNase inhibitor

[Takara, cat. number: 2313 A], 17U/µl Superscript II reverse transcriptase [Thermo Fisher Scientific, Invitrogen, cat. number: 18064071]) was added to each sample and the reaction took place as follows: 42 °C for 90 min, 10 cycles of 50 °C for 2 min, 42 °C for 2 min, followed by, 70 °C for 15 min and 4 °C on hold. For the pre-amplification PCR, 15 µl of PCR mastermix was added to each sample (1x Kapa HiFi HotStart ReadyMix [Roche, cat. number: KK2601], 0.1 µM forward primers [5'-TCGTCGGCAGCGTCAGATGTGTATAAGAGACAGA TTGCGCAATG-3'; IDT] and 0.1 µM reverse primers [5'-ACAGAGCATCA GCAGCATACGA-3', IDT]) and the reaction took place as follows: 98 °C for 3 min, 8 cycles of 98 °C for 20 sec, 67 °C for 15 sec, 72 °C for 6 min, and followed by 72 °C for 5 min, and 4 °C on hold.

**cDNA purification.** Purification of the cDNA libraries was performed by combining the cDNA samples to 22% PEG magnetic beads at a ratio of 1:0.8. Briefly, the mixture was incubated at room temperature for 8 min and on the magnetic rack for 5 min. The supernatant was removed, and the bead pellet was washed twice with freshly prepared 80% ethanol. The bead pellet was left to air-dry for 3 min, and the cDNA libraries were eluted in 17 µl of EB buffer [Qiagen, cat. number: 19086]. Quantification of the cDNA libraries was performed using the Quantifluor dsDNA kit [Promega, cat. number: E4871]. The libraries were normalised to 1 ng/ul.

**Tagmentation.** 2 ng of cDNA was combined with 18 µl of tagmentation mix containing 10 mM TAPS-NaOH [Sigma Aldrich, cat. numbers: T5130, S8045], 5 mM MgCl2 [Thermo Fisher Scientific, Invitrogen, cat. number: AM9530G], 8% PEG-8000 [Sigma Aldrich, cat. number: P2139] and 0.5 µl of in-house produced Tn5 at 44.5 µM. The samples were incubated at 55 °C for 8 min. To strip the Tn5 off the cDNA, 3 µl of 0.2% SDS [Sigma Aldrich, cat. number: L3771] solution was added to each sample, and the mixture was incubated at room temperature for 5 min.

**PCR amplification.** 1.5 µl of combined Nextera i7 and i5 [IDT] were added to each sample as well as 25 µl of PCR mastermix (1x KAPA HiFi PCR buffer, 0.06 mM (each) dNTPs [Thermo Scientific, cat. number: R0193], 1U KAPA HiFi polymerase [Roche, cat. number: KK2102]). The reaction took place as follows: 72 °C for 3 min, 95 °C for 30 sec, 10 cycles of 95 °C for 10 sec, 55 °C for 30 sec, 72 °C for 30 sec followed by 72 °C for 5 min and 4 °C on hold. The libraries were pooled and purified as described above. The final concentration of the pool was measured on a Qubit 4.0 using the Qubit dsDNA High Sensitivity Assay kit [Thermo Fisher Scientific, Invitrogen, cat. number: Q33230] and the library fragment distribution was inspected on an Agilent 2100 Bioanalyzer using Agilent High Sensitivity DNA chips. The library pool was sequenced on a Nextera Nextseq 550 using a Nextseq 500/550 High-Output 75 cycle sequencing kit v2.5 [Illumina #20024906] with the following settings: Read 1 = 72 cycles, Index 1 = 10 cycles, Index 2 = 10 cycles.

## RNA-seq data quantification

Raw multiplexed RNA-seq data were aligned and quantified to the GRCg6a genome (Gallus_gallus.GRCg6a.dna_sm.toplevel.fa.gz + Gallus_gallus.GRCg6a.100.gtf transcript annotations) using zUMIs[60] (v.2.9.4c, UMI(12-19), find_pattern: ATTGCGCAATG, additional_STAR_params: '--clip3pAdapterSeq CTGTCTCTTATACACATCT') with barcode- and UMI filtering cutoffs allowing 1 base at phred 20 using a list of expected barcodes for edit distance-based binning.

**Variant calling and allelic quantification.** For demultiplexed and aligned bam files, read groups were added and samples were merged according to genotype using GATK[61] (v4.1.3.0, AddOrReplaceReadGroups, MergeSamFiles [gatk --java-options "-Xmx128G"]). Variants were called using bcftools[62] (v.1.10.2) mpileup (--max-depth 8000 --skip-indels) and call (-mv, in ploidy mode) then filtered for a depth

over 5 reads with an allele frequency >50% using bcftools filter (-i 'D$p$ > 5 & AF > 0.5 & QUAL > 10'). Next, "unique WL" variants were subsetted from RJF variants using bcftools isec (-C -w 1) and "common RJF" variants were subsetted from the "unique WL" list. As a 2nd pass filtering, allelic expression was quantified for "unique WL" variants and variant-level count tables were calculated using zUMIs with a standard GRCg6a reference genome (see below) and variants with an agreement with the chicken strain in <50% of males or females were excluded to form the "final WL" variant list. Next, a custom GRCg6a reference genome was created by first inserting "common RJF" variant bases to correct for strain deviations using bcftools consensus (v.1.10.2) followed by N-masking using "final WL" variants using bcftools consensus (v.1.10.2, --mask). A STAR index was created using the WL N-masked GRCg6a genome and used for zUMIs alignment and quantification. Allelic quantification was performed on the zUMIs output bam files as previously described[19,31]. Briefly, variants were mapped to transcriptome positions and intersected with bases overlapping N-masked positions of the genome using the CIGAR string and reads were assigned to RJF/WL genotypes if >0.66 of basecalls matched the genotype and allelic read counts were summarised per gene and cell.

### CEF Truseq bulk RNA-seq

**Library preparation.** Bulk RNA-sequencing using Illumina's TruSeq RNA Library Prep Kit v2 [Illumina, cat. number: RS-122-2001] was performed in triplicate using 500 ng of purified RNA per sample according to the manufacturer's instructions. Briefly, the samples were incubated at 65 °C for 5 min, followed by bead purification to separate and elute polyA RNA-bound beads. The eluted RNA was incubated at 94 °C for 8 min followed by 4 °C on hold to elute, fragment, and prime the RNA for first-strand synthesis. First-strand synthesis was performed by adding 8 μl of first-strand master mix (containing 1ul of Superscript II reverse transcriptase for each 9 μl First Strand Master mix [Thermo Fisher Scientific, Invitrogen, cat. number: 18064071]) to each sample and running the following program: 25 °C for 10 min, 42 °C for 50 min, 70 °C for 15 min and hold at 4 °C. For second-strand synthesis, 25 μl of second-strand master mix was added to each sample, followed by incubation at 16 °C for 1h. After AMPure XP bead purification [Beckman-Coulter, cat. number: A63881], end repair was performed by adding 40 μl End Repair Mix to each sample and incubating at 30 °C for 30 min. After bead purification, the 3' ends were adenylated by adding 12.5 μl A-tailing mix to each sample followed by incubation at 37 °C for 30 min, 70 °C for 5 min and hold at 4 °C. Indexing adapters were ligated by adding 2.5 μl Ligation mix and 2.5 μl RNA adaptor Index (unique to each sample) per sample, and the mixture was incubated at 30 °C for 10 min, followed by the addition of 5 μl of Stop Ligation buffer to each sample to stop the ligation reaction. After bead purification, DNA fragments were enriched through PCR by adding 5 μl PCR primer cocktail and 25 μl PCR Master Mix to each sample. The reaction was performed using the following program: 98 °C for 30 s, 15 cycles of 98 °C for 10 s, 60 °C for 30 s, 72 °C for 30 s followed by 72 °C for 5 min and on hold at 10 °C. Following a final AMPure XP [Beckman-Coulter, cat. number: A63881] bead purification, the library quality and size were validated by running the samples on a High Sensitivity dsDNA Bioanalyzer chip and quantified by real-time quantitative PCR (RT-qPCR). Libraries were pooled in equimolar amounts and sequenced on a Nextseq 550 instrument using a Nextseq 500/550 High-Output 75 cycle sequencing kit v2.5 [Illumina #20024906] with the following settings: Read 1 = 76 cycles, Index 1 = 6 cycles, Index 2 = 6 cycles.

**Data analysis.** Raw BCL files were converted to FASTQ and demultiplexed using bcl2fastq (v.2.20.0.422). The demultiplexed raw data was aligned and quantified to an N-masked GRCg6a genome (see section *Variant Calling and allele quantification* under *Bulk RNA-sequencing using UMI-Smart-seq2*) and transcriptome (Gallus_gallus.GRCg6a.

100.gtf transcript annotations) using zUMIs (v.2.9.4c, cDNA (1-76), BC(1-8), additional_STAR_params: '--clip3pAdapterSeq AGATCGGAAGAGCA-CACGTCTGAACTCCAGTCA') with barcode filtering cutoffs allowing 1 base at phred 20 using a list of expected barcodes for edit distance-based binning. Allele calling was performed as described in section *Variant Calling and allele quantification* under *Bulk RNA-sequencing using UMI-Smart-seq2*.

### Single-cell RNA-sequencing using Smart-seq3

**Library preparation.** All scRNA-seq libraries were prepared as previously described[38]. Briefly, chicken embryonic fibroblasts (CEFs) were sorted into 384-well PCR plates containing 3 μl of lysis buffer (5% PEG-8000 [Sigma Aldrich, cat. number: P2139], 0.1% Triton-X-100 [Sigma Aldrich, cat. number: T8787], 0.5 units/μl RNase Inhibitor [Takara, cat. number: 2313 A], 0.5 mM (each) dNTPs [Thermo Fisher Scientific, cat. number: R0193], 1 μM oligo-dT primer [5'-Biotin-ACGAGCATCAG-CAGCATACGAT30VN-3'; IDT]. Sorting was performed using a FACS Aria II. After sorting, the plates were sealed, briefly centrifuged, and stored in −80 °C. To ensure cell lysis and RNA denaturation, the plates were incubated at 72 °C for 10 min and immediately placed on ice. For reverse transcription, 1 μl of reverse transcription master mix (25 mM Tris-HCl pH 8.3 [Sigma], 30 mM NaCl [Ambion; Thermo Fisher Scientific], 2.5 mM MgCl2 [Thermo Fisher Scientific, Invitrogen, cat. number: AM9530G], 1 mM GTP [Thermo Scientific, cat. number: R1461], 8 mM DTT [Thermo Fisher Scientific, cat. number: 707265 ML], 0.5 units/μl RNAse Inhibitor [Takara, cat. number: RR047A], 2 μM template-switching oligo [5'-biotin-AGAGACAGATTGCGCAATGNNNNNNN NrGrGrG-3'; IDT], 2U/μl Maxima H- RT enzyme [Thermo Scientific, cat. number: K1682]) was added to each sample. Reverse transcription was performed at 42 °C for 90 min, followed by 10 cycles of 50 °C for 2 min and 42 °C for 2 min, and terminated at 85 °C for 5 min. For PCR pre-amplification, 6 μl of PCR master mix (1x KAPA HiFi HotStart Buffer [Roche, cat. number: KK2502], 0.3 mM (each) dNTPs [Thermo Fisher Scientific, cat. number: R0182], 0.5 mM MgCl2 [Thermo Fisher Scientific, Invitrogen, cat. number: AM9530G], 0.5 μM forward primer [5'-TCGTCGGCAGCGTCAGATGTGTATAAGAGACAGATTGCGCAA-3'; IDT], 0.1 μM reverse primer [5'-ACGAGCATCAGCAGCATAC*G*A-3'; IDT], 0.02U/μl polymerase) was added to each sample. PCR pre-amplification was performed using the following thermocycler program: 98 °C for 3 min, 20 cycles of 98 °C for 20 sec, 65 °C for 30 sec, 72 °C for 4 min, followed by 72 °C for 5 min and 4 °C on hold. cDNA purification was performed using in-house prepared 22% PEG beads at a beads-to-sample ratio of 0.6:1. cDNA was quantified using the Quantifluor dsDNA kit [Promega, cat. number: E4871]. cDNA was normalised to a final concentration of 100 pg/μl. For the tagmentation step, 100 pg of cDNA were incubated with 1 μl of tagmentation mastermix (0.1 μl of tagmentation buffer 4x containing 40 mM Tris-HCl pH 7.5, 20 mM MgCl2 [Thermo Fisher Scientific, Invitrogen, cat. number: AM9530G], 20% Dimethylformamide [Sigma Aldrich, cat. number: D4551, 0.1 μl Amplicon Tagment Mix - Tn5 [Illumina, cat. number: FC-131-1096], 0.40 μl water) at 55 °C for 10 min. To strip the Tn5 from the cDNA, 0.5 μl of freshly prepared 0.2% SDS solution [Sigma Aldrich, cat. number: L3771] was added to each sample and incubated at room temperature for 5 min. The samples were indexed using 1 μl of 1 μM in-house, pre-mixed Nextera index primers [IDT] and post-tagmentation PCR was performed by adding 3 μl of PCR mastermix (1.4 μl Phusion HF 5x buffer [Thermo Fisher Scientific, cat. number: F530L], 0.2 mM (each) dNTPs [Thermo Fisher Scientific, cat. number: R0182], 0.01U/μl Phusion HF polymerase [Thermo Fisher Scientific, cat. number: F530L]) to each sample. PCR was performed using the following program: 72 °C for 3 min, 98 °C for 3 min, 10 cycles of 98 °C for 10 sec, 55 °C for 30 sec, 72 °C for 30 sec, followed by 72 °C for 5 min and 4 °C on hold. The samples were subsequently pooled and purified using in-house 22% PEG magnetic beads with a ratio of beads-to-sample of 0.7:1.

## Single-cell RNA-seq using UMI spike-ins

**Library preparation.** Full-length single-cell RNA-seq library preparation using the Xpress-seq (v1) method was performed at Xpress Genomics (Stockholm, Sweden). In brief, single cells were sorted using a Sony SH800S instrument into provided 384-well plates containing lysis buffer, spun down and stored at −80 °C. Upon submitting plates to Xpress Genomics, robotic automated library preparation was performed. Sequencing was performed on the DNBSEQ G400RS platform (MGI Tech) using App-C Sequencing primers.

**Data analysis.** Data was pre-processed using zUMIs (v.2.9.7) as described above, but with the following modifications: 2 mismatches were allowed in the detection of UMI-read patterns, and for barcode and UMIs, 4 and 3 mismatches were allowed, respectively. Additionally, spike-in sequences for the 5′ complex set of molecular spikes were included as mappable sequences (https://raw.githubusercontent.com/sandberg-lab/molecularSpikes/main/fasta_reference/molecularSpikes_complexset_5p.fa). Molecular spikes were extracted from aligned bam files using the UMIcountR package[43] (https://github.com/cziegenhain/UMIcountR) and overrepresented spike-ins were removed (>5 barcode or >100 sequences). Next, cells with less than 10% reads in spike-ins were kept and outliers were detected based on low gene detection (log 3 MADs) or read counts (log 5 MADs) and excluded. Spike-in size factors were calculated and UMIs were normalised using scater/scran (v.1.24.0, computeSpikeFactors, logNormCounts transform = "none").

## Public DNA methylation data analysis

Raw whole-genome bisulfite sequencing data for white leghorn samples was obtained from SRR1258373, SRR1258374, SRR1258375 and SRR1258376. Data was quality- and adaptor trimmed and low complexity reads were excluded using fastp (--low_complexity_filter −detect_adapter_for_pe) and aligned to the GRCg6a genome using abismal[63] (v.3.2.2, default settings). Sex was determined by calculating read coverage of chromosome W. 5mC methylation levels was calculated for CpG sites only and symmetrical CpG methylation counts was summarised using dnmtools[64] (v.1.4.2, format, counts -cpg-only, sym). Methylation counts were merged per sex using dnmtools merge and summarised using dnmtools merge (-t). CpGs with at least 5 total read counts in both males and females were kept. For genome-wide visualisation, methylated- and total counts were aggregated into 10 kb bins where bins within 1 Mb of centromeres (obtained from the UCSC table browser gap track) were excluded. Methylation fractions were calculated as methylated counts / total counts.

## Genome-wide mappability calculation

The GRCg6a genome (Gallus_gallus.GRCg6a.dna_sm.toplevel.fa) was indexed and k50-mer mappability was calculated using genmap[65] (v.1.3.0, map -K50 -E 0). Mappability was rounded to two decimals using awk (v.5.1.0, '{OFS = "\t";$4=sprintf("%.2 f",$4)}1'), coordinate-sorted then converted to bigwig format using the UCSC tool bedGraphToBigWig (v.4). To identify contiguous regions of low mappability, sliding windows were created across the genome using bedtools (v.2.30.0, makewindows -w 5000 -s 1000) and mappability per window was calculated using deeptools (v.3.5.4.post1, multiBigwigSummary BED-file). Next, windows with <80% mappability were extracted using awk and windows within 10 kb distance were merged using bedtools (v.2.30.0, merge -d 10000). Regions of >10 kb were kept and used for masking low mappability regions (called 'lowmap' below) where indicated.

## DNA-sequencing

**Library preparation.** Genomic DNA was isolated from cultured CEFs using the Monarch Genomic DNA purification kit according to the manufacturer's instructions. The concentration of gDNA was quantified using a Nanodrop 2000 instrument and the gDNA was subsequently diluted to 1 ng/µl. For DNA-sequencing, gDNA tagmentation was performed using an in-house prepared Tn5 enzyme as previously described[66]. Briefly, 5 ng of gDNA was incubated with 15 µl of tagmentation mastermix (10 mM TAPS [Sigma], 5 mM MgCl2 [Thermo Fisher Scientific, Invitrogen, cat. number: AM9530G], 10% Dimethylformamide [Sigma Aldrich, cat. number: D4551], 2.25 µM Tn5 [in-house]) at 55 °C for 8 min. To strip the Tn5 from the DNA, 3.5 µl of 0.2% SDS [Sigma Aldrich, cat. number: L3771] was added to each reaction. The samples were quickly centrifuged and incubated at room temperature for 5 min. The samples were indexed using 2.5 µl of 1 µM pre-mixed Nextera index primers [IDT]. Post-tagmentation PCR was performed by adding 16.5 µl of PCR master mix (1x KAPA HiFi PCR buffer, 0.6 mM (each) dNTPs, 1U/µl KAPA HiFi polymerase [Roche, cat. number: KK2102]) to each sample and incubating using the following program: 72 °C for 3 min, 95 °C for 30 sec, 6 cycles of 95 °C for 10 sec, 55 °C for 30 sec, 72 °C for 30 sec, followed by 72 °C for 5 min and 4 °C on hold. Double purification of the final libraries was performed using in-house 22% PEG magnetic beads[38]. Briefly, 22% PEG magnetic beads were combined with the pooled DNA libraries in a bead-to-sample ratio of 0.9:1 and incubated at room temperature for 8 min. The samples were then placed on a magnetic rack for 5 min. The clear supernatant was then removed and discarded, and the bead pellets were washed twice with freshly prepared 80% EtOH. The beads were left to air-dry for 3 min while remaining on the magnetic rack. The samples were eluted in 30 µl. To ensure complete removal of residual impurities and primer-dimers, the purification was repeated as described above and the final sample was eluted in 17 µl of nuclease-free water [Ambion]. Library fragment size was assessed using a Bioanalyzer high-sensitivity dsDNA chip and library concentrations were quantified using Qubit's high-sensitivity dsDNA quantification kit on a Qubit 3.0 Fluorometer. Libraries were pooled in equimolar amounts and sequenced on a Nextseq 550 instrument using a Nextseq 500/550 High-Output 75 cycle sequencing kit v2.5 [Illumina #20024906] with the following settings: Read 1 = 74 cycles, Read 2 = 74 cycles, Index 1 = 10 cycles, Index 2 = 10 cycles.

**DNA analysis.** Raw DNA-seq data was adaptor- and quality trimmed using fastp[67] (v.0.20.0, −adapter_sequence CTGTCTCTTATACACATCT −adapter_sequence_r2 CTGTCTCTTATACACATCT) and aligned to the GRCg6a reference genome using minimap2[68] (v.2.24-r1122, -ax sr). Reads were sorted, mate-pair information fixed, and duplicates marked using biobambam2 (v.2.0.87, bamsort fixmates=1 markduplicates=1). Variants were called using bcftools mpileup (−ignore-RG -a AD,DP, −max.depth 8000) and call (-mv, in ploidy mode) using ZZ and ZW ploidies with sample-sex information. Variants within 5 bp of indels were excluded and heterozygous variants sequenced to a depth over 5 reads with a minor allele frequency over 10% were filtered using bcftools filter (-g 5 -i 'TYPE = "snp" & QUAL > 10 & INFO/Dp > 5 & GT = "het" & MAF > 0.1'). To calculate DNA copy numbers, the GRCg6a genome was binned into 100 kb bins using bedtools[69] makewindows (v.2.30.0, -w 100000) and binned read counts were calculated using bedtools multicov (-q 13). Genome statistics were also calculated for the same bins; mappability (see above) using deeptools (multiBigwigSummary BED-file); nucleotide frequencies using bedtools (nuc); assembly gaps were obtained from UCSC and gaps >1 kb were kept and bins within 500 kb were identified using bedtools (window -w 500000 -c); RepeatMasker rmsk track was obtained from UCSC and overlapped with bins using bedtools (intersect -wao | map -c 10) and percentage overlap was calculated. Bins with >2.5% N bases or average mappability <50% or within 500 kb of a large assembly gap or with a rmsk fraction 2 MADs above median were excluded. Data was corrected for GC-content and mappability and DNA copies were estimated using HMMcopy (v.1.38.0, correctReadcount mappability = 0.8). Expected ploidies were set to

2 for diploid samples and 3 for triploid and multiplied with DNA copies to adjust for ploidy and regions annotated as ideal by HMMcopy were used for plotting. For base-resolution variants, only variants with a read depth of 6-50, heterozygous genotype and >0 variance were kept and variants overlapping excluded genome bins were removed.

## ATAC-seq

**Library preparation.** Omni ATAC-seq libraries were prepared as follows: Briefly, 100 000 cells were pelleted at 500 x g at 4 °C for 5 min. The supernatant was carefully removed, and the pellet was resuspended in 50 μl ATAC-RSB lysis buffer (10 mM Tris-HCl pH 7.4, 10 mM NaCl, 3 mM MgCl2 [Thermo Fisher Scientific, Invitrogen, cat. number: AM9530G], 0.1% NP-40, 0.1% Tween-20, 0.01% Digitonin) and incubated on ice for 10 min. The lysis buffer was washed out by adding 1 ml of ATAC-RSB wash buffer containing 0.1% Tween but no NP-40 or digitonin and the samples inverted 3 times to mix, and the nuclei pelleted at 500 x $g$ at 4 °C for 10 min. The supernatant was removed, and the samples were resuspended in 50 μl transposition mixture (25 μl 2x TD buffer, 1.5 μl Tn5 (27 μM) 16.5 μl 1x PBS, 0.5 μl 1% digitonin, 0.5 μl 10% Tween-20, 7 μl water). Tagmentation was performed in a thermoshaker at 37 °C for 30 min at 1000 rpm. The transposed DNA was purified using the Zymo DNA Clean and Concentrator-5 kit following the manufacturer's instructions and eluted in 21 μl of elution buffer. Library amplification was performed by adding 30 μl of PCR master mix to the purified DNA (25 μl 2x NEBNext High Fidelity PCR Master Mix, 2.5 μl Ad1_noMX (common i5 Nextera adaptor primer), 2.5 μl Ad2 (unique i7 Nextera adaptor primer). The libraries were amplified for 11 cycles using the following cycling conditions: 72 °C for 5 min, 98 °C for 30 s, 11x (98 °C for 10 s, 63 °C for 30 s, 72 °C for 1 min), 4 °C on hold. The final libraries were fragment size-selected by double-sided 0.5x/1.3x bead purification using homemade 22% PEG magnetic beads. Briefly, 25 μl of room temperature Ampure XP beads [Beckman-Coulter, cat. number: A63881] were added to each sample (beads-to-sample ratio = 0.5) and incubated for 10 min after thorough resuspension. The samples were placed on a magnetic rack and the supernatant was removed and transferred to a new microcentrifuge tube containing 65 μl room temperature Ampure XP beads [Beckman-Coulter, cat. number: A63881] (beads-to-sample (original volume) ratio = 1.3). After thorough mixing, the samples were incubated at room temperature for 10 min and placed on a magnetic rack for 5 min. The supernatant was discarded, and the beads were washed twice with 200 μl of freshly prepared 80% ethanol. After ethanol removal, the samples were air-dried for 5 min and the libraries eluted in 20 μl of nuclease-free water. The libraries were pooled in equimolar amounts and paired-end sequencing was performed on an Illumina Nextseq 550 instrument to obtain ~20 million paired-end reads per sample.

**Data preprocessing.** Raw ATAC-seq data was converted to FASTQ format using bcl2fastq (v.2.20.0.422). Sequencing adaptor removal was performed using fastp (v.0.20.0). The resulting trimmed reads were aligned to the GRCg6a genome using Bowtie 2 (v.2.5.1, settings: -N 1, -X 2000, --very-sensitive). Aligned reads were then converted to BAM format, mate pair information was fixed, duplicate read pairs were marked, and the BAM output was sorted according to read coordinates using biobambam2 (v.2.0.87, bamsort fixmates=1, markduplicates=1). Replicate merging was performed using sambamba merge (v. 0.7.0). SNPs were called from the ATAC-seq data using bcftools (v. 1.10.2, mpileup --ignore-RG -Ou -a AD,DP --max-depth 8000 | call --threads 64 -mv -Oz --ploidy-file –samples-file | filter -g 5 -i 'TYPE = "snp" & QUAL > 10 & INFO/D$p$ > 5 & GT = "het" & MAF > 0.1'). Next, the GRCg6a genome was N-masked for SNP positions using bcftools (consensus --mask) and data was realigned to the N-masked reference using bowtie2 as described

above. Peak calling on all replicates was performed using Genrich in ATAC-seq mode (v.0.6, settings: -j, -r, -d 150, -e W,MT, -E lowmap), PCR duplicates were removed, the cut sites were expanded to 150 bp and reads from the W chromosome and the mitochondria were excluded, and it was repeated both with and without low mappability regions. Genrich requires input sorted by query name which was done by bio-bambam2 bamsort. All peaks were combined and replicated peaks within 100 bp were merged and counted using bedtools (v.2.30.0, merge -c 1 -o count -d 100) and peaks present with >1 counts were kept. Distance to nearest TSS for each peak was annotated using bedtools (v.2.30.0, closest). Peak quantification of proper read pairs, both excluding and including low-mapping regions, and removing PCR duplicates was calculated by deepTools (v.3.5.4.post1, multi-BamSummary --samFlagInclude 2, --ignoreDuplicates, -bl lowmap). Genome-wide signal pileups was normalised using deepTools (bam-Coverage --normalizeUsing RPGC, --effectiveGenomeSize 1058535536, --ignoreDuplicates, --samFlagInclude 2, -ignore MT, --minFragmentLength 38, --maxFragmentLength 2000). These only included proper read pairs, mitochondrial entries were removed as well as PCR duplicates, and only fragments between 38 and 2000 bp lengths were kept. The pileups were also made by containing, and excluding, low mapping regions, for further use and visualisations. Effective genome size of GRCg6a used for normalisation was obtained from Genrich. Enrichment 5 kb around TSS was calculated using deeptools (computeMatrix -a 5000, -b 5000 -R Gallus_gallus.GRCg6a.100.gtf).

**Allelic analysis.** To phase the Z chromosome, all SNP positions were extracted using bcftools corresponding to all SNPs ('all_snps'; query -f '%ID\t%CHROM\t%POS\t1\t%REF/%ALT\n') and chrZ SNPs with a genotype matching the non-present allele in either female sample ('female_gt_snps'; query -i 'CHROM = "Z" & & (GT[0] = "0" | GT[1] = "1") ' -f '%ID\t%CHROM\t%POS\t1\t%ALT/%REF\n'). Next, REF and ALT bases were flipped for SNPs not matching genotypes in females using awk ('NR = = FNR{a[$2,$3] = $5;next}(($2,$3) in a){OFS = "\t";$5=a[$2,$3]}1' female_gt_snps all_snps). These corrected SNPs were used as input to SNPsplit (v.0.4.0, –no_sort –snp_file corrected_snps) to split the data into respective alleles. Peak calling, peak quantification and TSS enrichment were performed as described above for each allele.

**Transcription factor footprinting and identification of differentially bound transcription factors.** Differential footprinting analysis was performed on called peaks (see section ATAC-seq *data analysis*) using TOBIAS (v0.16.1, default settings). ATACorrect was applied to correct for Tn5 bias and resolve hidden footprints. To identify regions of protein binding across the genome, footprinting scores were calculated across the open chromatin regions, using the corrected signals, with the TOBIAS ScoreBigwig. Differential TF binding analysis was performed with TOBIAS BINDetect module which combines footprinting scores with TF motif information from JASPAR CORE 2024 (non-redundant, vertebrate). The heatmap was generated using ComplexHeatmap and the motif clustering using motifStack.

**E-box motif enrichment.** To count E-box motifs per chromosome, chromosomal regions were extracted from the chicken genome (Gallus_gallus.GRCg6a.dna_sm.toplevel.fa) using samtools (v1.10) and appended to individual files. Known E-box motifs were extracted from Homer (v5.1) and motifs per chromosome were located using Homer's findMotif.pl script. The ratio of E-box motifs per non-N-masked base was calculated by dividing the total number of E-box motifs per chromosome by the length of the chromosome in non-N-masked bases. To calculate the observed/expected (O/E) enrichment ratio, each E-box motif per base ratio for each chromosome was divided by the ratio of number of genome-wide E-box motif/genome-wide length of non-N-masked bases.

## Proteomics

**Sample preparation.** Cell pellets of approximately 1 million cells were collected at 300 x g for 5 min and washed with ice-cold PBS 5 times to eliminate serum-containing media. Cell pellets were solubilized in 20 µl of 8 M urea in 50 mM Tris-HCl, pH 8.5 sonicated in water bath for 5 min before 10 µl of 1% ProteaseMAX surfactant (Promega) in 10% acetonitrile (ACN) and Tris-HCl as well as 1 µl of 100x protease inhibitor cocktail (Roche) was added. The samples were then sonicated using VibraCell probe (Sonics & Materials, Inc.) for 40 s with pulse 2-2 s (on/off) at 20% amplitude. Protein concentration was determined by BCA assay (Pierce) and a volume corresponding to 25 µg of protein of each sample was taken and supplemented with Tris-HCl buffer up to 90 µl. Proteins were reduced with 3.5 µl of 250 mM dithiothreitol in Tris-HCl buffer, incubated at 37 °C during 45 min and then alkylated with 5 µl of 500 mM iodoroacetamide at room temperature (RT) in dark for 30 min. Then 0.5 µg of sequencing-grade modified trypsin (Promega) was added to the samples and incubated for 16 h at 37 °C. The digestion was stopped with 5 µl cc. formic acid (FA), incubating the solutions at RT for 5 min. The sample was cleaned on a C18 Hypersep plate with 40 µl bed volume (Thermo Fisher Scientific), dried using a vacuum concentrator (Eppendorf). Peptides, equivalent of 25 µg protein, were dissolved in 70 µl of 50 mM triethylammonium bicarbonate (TEAB), pH 7.1 and labelled with TMTpro mass tag reagent kit (Thermo Fisher Scientific) adding 100 µg reagent in 30 µl anhydrous ACN in a scrambled order and incubated at RT for 2 h. The reaction was stopped by addition of hydroxylamine to a concentration of 0.5% and incubation at RT for 15 min before samples were combined and cleaned on a C-18 HyperSep plate with 40 µl bed volume. The combined TMT-labelled biological replicates were fractionated by high-pH reversed-phase after dissolving in 50 µl of 20 mM ammonium hydroxide and were loaded onto an Acquity bridged ethyl hybrid C18 UPLC column (2.1 mm inner diameter × 150 mm, 1.7 µm particle size, Waters), and profiled with a linear gradient of 5–60% 20 mM ammonium hydroxide in ACN (pH 9.0) over 48 min, at a flow rate of 200 µl /min. The chromatographic performance was monitored with a UV detector (Ultimate 3000 UPLC, Thermo Scientific) at 214 nm. Fractions were collected at 30 s intervals into a 96-well plate and combined in 12 samples concatenating 8-8 fractions representing peak peptide elution.

**Liquid chromatography-tandem mass spectrometry data acquisition.** The peptide fractions in solvent A (0.1% FA in 2% ACN) were separated on a 50 cm long EASY-Spray C18 column (Thermo Fisher Scientific) connected to an Ultimate 3000 nano-HPLC (ThermoFisher Scientific) using a gradient from 2-26% of solvent B (98% AcN, 0.1% FA) in 90 min and up to 95% of solvent B in 5 min at a flow rate of 300 nL/min. Mass spectra were acquired on an Orbitrap Fusion Lumos tribrid mass spectrometer (Thermo Fisher Scientific) ranging from m/z 375 to 1500 at a resolution of $R = 120,000$ (at m/z 200) targeting $4 \times 10^5$ ions for maximum injection time of 50 ms, followed by data-dependent higher-energy collisional dissociation (HCD) fragmentations of precursor ions with a charge state 2+ to 6 +, using 45 s dynamic exclusion. The tandem mass spectra of the top precursor ions were acquired in 3 s cycle time with a resolution of $R = 50,000$, targeting $1 \times 10^5$ ions for maximum injection time of 150 ms, setting quadrupole isolation width to 0.7 Th and normalised collision energy to 35%.

**Data analysis.** Acquired raw data files were analyzed using Proteome Discoverer v3.0 (Thermo Fisher Scientific) with MS Amanda v2.0 search engine against Gallus gallus protein database (UniProt). A maximum of two missed cleavage sites were allowed for full tryptic digestion, while setting the precursor and the fragment ion mass tolerance to 10 ppm and 0.02 Da, respectively. Carbamidomethylation of cysteine was specified as a fixed modification. Oxidation on methionine, deamidation of asparagine and glutamine, as well as acetylation of N-termini and TMTpro were set as dynamic modifications. Initial search results were filtered with 1% FDR using the Percolator node in Proteome Discoverer. Quantification was based on the reporter ion intensities

## Ribosomal profiling

### Library preparation

**Isolation of ribosome-protected fragments (RPFs).** Cells were grown to 80 % confluency on 2 × 15-cm dishes. Medium was discarded and plates were shortly submerged in liquid nitrogen to snap freeze cells. 300 µl of 2x lysis buffer (50 mM Tris pH 7.5, 200 mM NaCl, 20 mM MgCl₂, 2 mM DTT, 200 µg/ml cyclohexamide, 2 % Triton X-100, 2x Complete EDTA-free protease inhibitor cocktail [Roche, cat. number: 04693132001], 4000 U/ml TURBO DNase I [Thermo Fisher Scientific, Invitrogen, cat. number: AM2238]) was added dropwise on each plate and lysates were collected using cell scrapers. Cell debris were removed by centrifugation (10,000 x g, 15 min, 4 °C). RNA concentrations were measured by Qubit RNA Broad Range kit [Thermo Fisher Scientific, Invitrogen, cat. number: Q10210] and 90 µg were subjected to RNase treatment for 45 min at 22 °C (750 U, Ambion RNase I [Thermo Fisher Scientific, Invitrogen, cat. number: AM2294]). RNase treatment was stopped by the addition of 15 µl RNase inhibitor (1 U/µl, SUPERase-In, [Thermo Fisher Scientific, Invitrogen, cat.number: AM2696]), which was followed by a short centrifugation step to remove insoluble material (5,000 x g, 5 min). Supernatants were loaded on 1 M sucrose cushions (25 mM Tris pH 7.5, 100 mM NaCl, 10 mM MgCl₂, 1 mM DTT, 100 µg/ml cyclohexamide, 1x Complete EDTA-free protease inhibitor cocktail [Roche, cat. number: 04693132001], 40 U/ml RNase inhibitor [Thermo Fisher Scientific, Invitrogen, cat. number: AM2696] in 11 × 34 mm tubes (Beckman Coulter) and ribosomes were pelleted at 55.000 rpm (290 000 x *g*) for 3 h in a TLS-55 rotor (Beckman Coulter). Afterwards, supernatants were removed, and ribosomal pellets were resuspended in 1 ml TRIZOL reagent [Thermofisher, cat. number: 15596018]. RNA was isolated according to manufacturer's instructions. Isolated RNA was heated at 80 °C for 3 min, put on ice for 1 min, mixed with Gel Loading Buffer II (ThermoFisher) and loaded onto a 15 % Novex TBE-Urea gel [Thermo Fisher Scientific, Invitrogen, cat. number: EC6885BOX]. The gel was run in 1x TBE buffer at 100 V for ~2 h. After completion of the run the gel was stained with 1x SYBR Gold Nucleic Acid Gel Stain [Thermo Fisher Scientific, Invitrogen, cat. number: S11494] in 1x TBE. Nucleic acids were visualised and bands referring from 25-35 nt were excised. RNA was extracted from gel slices in 600 µl RNA extraction buffer (300mM NaOAc, pH 5.5, 1mM EDTA, 0.25 % SDS), rotating at 4°C overnight. The next day, RNA was precipitated by adding 1.8 ml ice-cold EtOH together with 4 µl GlycoBlue Coprecipitant [Thermo Fisher Scientific, Invitrogen, cat. number: AM9516] and subsequent storage at −80°C ON. Precipitated RNA was pelleted by centrifugation (5,000 x g, 10 min, 4 °C). Pellet was once washed with 1 ml EtOH, dried for ~5 min and resuspended in 15 µl 10 mM Tris pH 7.5 supplemented with 1 µl RNase inhibitor [Thermo Fisher Scientific, Invitrogen, cat. number: AM2694].

**Ligation of adaptors to RPFs.** Samples were heated at 80 °C for 2 min before placing on ice. Next, 3' phosphates were removed by T4 PNK treatment (1 µl T4 PNK (NEB, cat. number: M0201S) added) in 1x T4 PNK buffer (NEB, cat. number: M0201S) at 37 °C for 2 h. Reaction was stopped by heat inactivation (65 °C, 10 min). RNA was pelleted by addition of 70 µl water, 2 µl GlycoBlue Coprecipitant (Thermo Fisher Scientific, Invitrogen, cat. number: AM9516), 10 µl 1 M NaOAc and 300 µl EtOH and subsequent storage at −80°C. RNA was washed and dried as described earlier and finally resuspended in 7 µl 10 mM Tris pH 7.5 supplemented with 1 µl RNase inhibitor. RNA libraries were generated using TruSeq Small RNA Library Prep Kit [Illumina, cat. number: RS-200-0012] according to the manufacturer's protocol with some

modifications. Preparation was started by adding 1.2 µl adenylated RA3 to dephosphorylated RNA and incubating the mixture at 80 °C for 2 min. Afterwards, ligation was performed by addition of 2 µl of T4 RNA Ligase 2 (truncated K227Q), 2 µl T4 RNA Ligase 2 buffer and 6 µl PEG8000 (all components from NEB) and incubation at 14 °C ON. RNA was precipitated as described earlier, 20 µl 3 M NaOAc and 600 µl EtOH) and resuspended in 4 µl 10 mM Tris pH 7.5. Ligation products were then purified on a 15 % Novex TBE-Urea gel [Thermo Fisher Scientific, Invitrogen, cat. number: EC6885BOX], extracted, and precipitated as described earlier. Next, RNA was resuspended in 13 µl 10 mM Tris pH 7.5 supplemented with 1 µl RNase inhibitor. Then, 2 mM ATP, 2 µl 10x T4 PNK buffer and 2 µl T4 PNK (NEB) were added, and the reaction mixture was incubated for 2 h at 37 °C, followed by heat inactivation (65 °C, 10 min). RNA was precipitated and resuspended in 13 µl 10 mM Tris pH 7.5 supplemented with 1 µl RNase inhibitor. Thereafter, RNA footprints were ligated with 5′ RNA adaptor (RA5, Illumina) by adding 1.2 µl RA5, 2 µl 10x T4 buffer and 2 µl T4 RNA ligase [Promega, cat. number: M1051] and incubating at 14 °C ON. RNA was precipitated and resuspended in 3 µl 10 mM Tris pH 7.5.

**Reverse transcription and PCR amplification of library.** Reverse transcription was performed using RNA RT primers from TruSeq Small RNA Library Prep Kit (Illumina) and SuperScript III First-Strand Synthesis System [Thermo Fisher Scientific, cat. number: 18080051] according to the manufacturer's protocol. Afterwards, 2 µl of RT products were PCR amplified using Phusion High-Fidelity PCR master mix [New England Biolabs, cat. number: M0541S] and DNA primers from TruSeq Small RNA Library Prep Kit [Illumina, cat. number: RS-200-0012]. The PCR products were resolved on a 10 % Novex non-denaturing TBE gel [Thermo Fisher Scientific, cat. number: EC6275BOX] using 1x TBE running buffer. PCR products were excised and extracted using DNA extraction buffer (300 mM NaCl, 10 mM Tris pH 8, 1 mM EDTA). Subsequently, PCR products were precipitated and pelleted. Libraries were resuspended in 12 µl 10 mM Tris pH 7.5.

**Duplex-specific nuclease (DSN) digestion.** To reduce the amount of ribosomal RNA contamination DSN digestion was performed using a DSN kit [Evrogen, cat. number: EA003]. First, 4 µl of hybridisation buffer (200 mM HEPES pH 7.5, 2 M NaCl) was added to the libraries. Next, libraries were heated for 2 min at 98 °C followed by incubation for 5 h at 68 °C. Consecutively, 1x master buffer (evrogen) together with 2 µl DSN enzyme were added to the samples and incubated additional 25 min at 68 °C. Digestion was stopped by addition of 20 µl stop solution (evrogen) and 5 min incubation at 68 °C. Finally, samples were cooled down on ice and DNA was isolated by phenol/chloroform extraction. Therefore, samples were mixed with 160 µl water and 200 µl phenol/chloroform (1:1) and the aqueous phase was precipitated as before. 2 µl of digested libraries were subjected to another round of PCR amplification and consecutive gel purification. Final libraries were resuspended in 11 µl 10 mM Tris pH 7.5. To assess fragment size distribution and final library concentrations, libraries were run on a Bioanalyzer instrument using a High Sensitivity dsDNA kit and quantified using the Qubit 1x dsDNA high sensitivity kit. Libraries were then pooled in equimolar amounts. Paired-end sequencing was performed on a Nextseq550 instrument to obtain ~20 million reads per sample using the following settings: Read 1 = 75 cycles, Index 1 = 6 cycles, Read 2 = 75 cycles.

**Data analysis.** Raw Ribo-seq BCL data was converted to FASTQ and demultiplexed using bcl2fastq (v. 2.20.0.422) followed by adaptor trimming using FASTP. For pre-alignment to ribosomal RNA (rRNA) sequences, a fasta file of rRNA sequences obtained from SILVA (release 138, smr_v4.3_default_db) was indexed using bowtie2-build. Pre-alignment to rRNA sequences was performed using bowtie2 (v. 2.5.1, settings: -N 1, --very-sensitive, --al-conc, --un-conc). Unaligned bowtie2 output aligned to the GRCg6a genome and transcriptome (v.2.7.2a, genome: 2.7.2a, transcriptome: Gallus_gallus.GRCg6a.100.gtf using STAR (v.2.7.2a, --runMode alignReads, --sjdbOverhang 31, --seedSearchStarLmax 10, --outFilterMultimapNmax 2, --quantMode TranscriptomeSAM). To ensure comparability between samples, 3.5 million reads were downsampled for each sample prior to P-site assignment and downstream analyses.

Downstream analysis including P-site assignment and transcript-level quantification was performed using Ribowaltz (v.2.0). PCR duplicates were removed with the option duplicates_filter (extremity = "both") and p-site offsets were calculated with default settings (flanking=6, extremity = "auto"). The per-gene sums of ribosome fragment counts mapping to the coding sequence (CDS) of protein-coding transcripts were used to calculate FPKM-normalised Ribo-seq counts, which were used in downstream analyses. For all paired Ribo-seq/RNA-seq analyses, bulk RNA-seq data (Truseq) was used, the translation index was calculated as Ribo-FPKM/RNA-FPKM per gene and sample. Similarly, Female:Male ratios were calculated as the mean translation index per gene and chromosome, including 1-33 and Z.

## Multiplexed Quantitative ChIP-seq
**Library preparation.** Triplicate pellets of $10^6$ cells were collected for all conditions, flash frozen and stored at −80 °C before use. Immunoprecipitation was performed using the EpiFinder Genome kit (Epigenica, EpGe001) according to the manufacturer's instructions. In brief, native frozen cell pellets were lysed and MNase digested to mono- to tri-nucleosome fragments and ligated with double-stranded DNA adaptors in a one-pot reaction. Barcoded samples were then pooled and aliquoted into individual ChIP reactions with Protein A [Thermo Fisher Scientific, cat. number: 10001D] for the following antibodies: H3K4me3 [Millipore; 04-745], H3K27ac [Active Motif; 39034], H3K9ac [Active Motif; 39137-AF], H4K16ac [Millipore; 07-329]. Briefly, 260 µl of sample pool was treated with 1 µl RNase A and 3 µl Proteinase K, followed by digestion at 37 °C for 15 min and at 63 °C for 45 min with agitation at 1000 rpm. AMPure XP bead [Beckman-Coulter, cat. number: A63881] purification was performed at 1:1 ratio of sample:beads with 2 × 80% EtOH washes, and the sample was eluted in 10 µl of elution buffer. Upon incubation overnight with rotation at 4 °C and washing steps, ChIP DNA was isolated and set up in sequential reactions of adaptor fill-in, in vitro transcription, RNA 3′ adaptor ligation, reverse transcription and PCR amplification to generate final libraries for each ChIP. After quality assessment and concentration estimation, libraries were combined and sequenced on an MGI DNBSEQ-G400RS instrument platform with paired-end settings.

**Data analysis.** Raw ChIP-seq data was converted to FASTQ format and demultiplexed using mgikit (v. 0.1.4; settings: -m 1), allowing for up to one mismatch in the barcodes. Each FASTQ file of the demultiplexed samples were concatenated across the four lanes, creating the sample-specific FASTQ files for further processing. Quality evaluation, mapping, scaling the data to input, and the creation of bigWig files was done using the minute workflow[70] (v. 0.6.0; settings: fragment_size: 400, max_barcode_errors: 1, mapping_quality: 0). The mapping was done using the GRCg6a reference genome, and low mapping regions were excluded. TSS enrichment was calculated using deepTools (computeMatrix reference-point -a 5000 -b 5000 --nanAfterEnd --skipZeros) for GRCg6a.100.gtf transcripts. The scaled and pooled sample bigWig files for the modifications H3K27ac, H3K4me3, H3K9ac and H4K16ac, as well as the merged and low mapping regions excluded bigWig files from the ATAC, were ran through ChromHMM-tools to create signal input for use in ChromHMM. The signal input obtained from ChromHMM-tools was binarized and HMM models were created with ChromHMM (v. 1.25) using six states, which was mapped to Galgal6a.

## Metaphase spreads and karyotyping of CEF samples

Metaphase spreads and karyotyping was performed based on the protocol by Howe and colleagues[71] as follows: Briefly, CEFs were grown as described above to a confluency of 70% and treated with CEF media containing 0.1 µg/ml colcemid [Gibco, cat. number: 15212012] for 80 min at 37 °C, 5% CO2. After 80 min, the treatment was removed, and the cells were washed with HBSS [Gibco, cat. number: 14175095] and detached using TryPLE [Gibco, cat. number: 12604021]. The cell suspension was then collected in media and centrifuged at 200 x $g$ for 10 min at room temperature. The supernatant was removed, leaving 0.5 ml in which the cell pellet was gently resuspended. For cell swelling, 10 ml of freshly prepared pre-warmed 0.075 M KCl [Sigma Aldrich, cat. number: 529552] solution was added to the resuspended cells drop-wise and the samples were incubated at 37 °C for 12 min with gentle agitation every 2 min. The samples were then centrifuged at 200 x $g$ for 5 min at room temperature. The supernatant was removed and discarded leaving 0.5 ml in which the cell pellet was gently resuspended. To fix the cells, 5 ml of freshly prepared Carnoy's fixative (3:1 ratio methanol:acetic acid [VWR, cat. numbers: K977-1L, 0714-500 ML]) was added while vortexing, followed by an additional 5 ml added without vortexing. The samples were centrifuged at 200 x $g$ for 5 min at room temperature. The above-mentioned step was repeated by adding 5 ml of Carnoy's fixative without vortexing. For the final fixation, the supernatant was discarded, leaving 0.5 ml in which the pellet was gently resuspended and 5 ml of Carnoy's fixative was added, and the samples were stored at +4 °C until slide preparation.

**Slide preparation.** Microscope slides were first placed in a Coplin jar filled with absolute ethanol for 10 min and rinsed in distilled water to prepare them for metaphase spreading. The cells were resuspended in freshly prepared Carnoy's fixative and dropped onto the prepared microscope slides from a distance of 5 cm. The slide was fixed with a large drop of Carnoy's fixative and left to dry at room temperature. Freshly prepared Giemsa solution (3:1 ratio of Gurr buffer [Gibco, cat. number: 10582013]: Giemsa stain [Gibco, cat. number: 10092013]) was added to the slides for 10 min at room temperature. The slides were rinsed in distilled water and left to dry at room temperature. Coverslips were then mounted using Permount [Fisher Chemical, cat. number: SP15100] and slides observed under 40x or 100x brightfield microscope objectives.

## Statistics and data visualisation

All statistical tests were performed in R as two-tailed tests unless otherwise stated. Plots were made using ggplot2[72] (v.3.5.2) and all heatmaps were produced using ComplexHeatmap (v.2.14.0). Boxplots are presented as median, first and third quartiles, and 1.5x inter-quartile range (IQR). For median ± confidence interval plots, boot-strapped 95% confidence intervals ($n = 1000$) were calculated using the percentile method[73] using the R boot package (v. 1.3-31).

## Reporting summary

Further information on research design is available in the Nature Portfolio Reporting Summary linked to this article.

## Data availability

Raw and processed sequencing data is available through ArrayExpress and the European Nucleotide Archive (ENA) under accession codes E-MTAB-14443 and PRJEB79933 (bulk RNA-seq), E-MTAB-14470 and PRJEB80546 (scRNA-seq), E-MTAB-14391 and PRJEB79402 (DNA-seq), E-MTAB-14390 and PRJEB79410 (ATAC-seq), E-MTAB-14392 and PRJEB79404 (ChIP-seq) and E-MTAB-14393 and PRJEB79407 (Ribo-seq). Proteomics data is available through PRIDE under accession PXD054989 (MS2/MS3 proteomics). Previously published raw data is available at SRA (Sequence Read Archive) under accession codes: SRR1258373, SRR1258374, SRR1258375 and SRR1258376. Uncropped

micrographs of karyotyping images included in Supplementary Fig. 2 and real-time quantitative PCR (RT-qPCR) data for miR-2954 expression in chicken tissues and CEF lines generated in this study are provided in the Source Data. Source data are provided as a Source Data file. Source data are provided with this paper.

## Code availability

Code to reproduce this work is available at Github (https://github.com/reiniuslab/Z-upregulation)[74].

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

## Acknowledgements

We thank members of the Reinius lab for their input, Akos Vevgari from the Proteomics Biomedicum Core Facility for his input and technical support, and employees of Epigenica for early access and technical support related to EpiFinder. Protein identification and quantification were carried out at the Proteomics Biomedicum core facility at Karolinska Institutet and Xpress-seq library preparation was performed at Xpress Genomics, Stockholm, Sweden. Sequencing was performed at the Division of Medical Systems Bioengineering (MBB, Karolinska Institutet), Xpress Genomics, and the National Genomics Infrastructure, SciLifeLab Stockholm, Sweden. In accordance with the funding agencies' policy, the results of this study were continuously made publicly available in preprint form (https://doi.org/10.1101/2024.08.20.608780, first version online August 20, 2024, originally submitted to Nature). This study was made possible by grants from the Knut & Allice Wallenberg Foundation (2021.0142 and 2022.0146) to BR, the Swedish Research Council (2022-01620) to BR, KI SFO StratRegen 2021 to BR, and the Swedish Society for Medical Research (CG-22-0260) to BR and (PD20-0217) to AL.

## Author contributions

NP performed the CEF culturing and analyses, including FACS analyses, bulk and single-cell RNA-seq, karyotyping, RT-qPCRs, proteomics, analysed data, prepared figures, and wrote the manuscript. AL performed data pre-processing, QC and allelic analysis of ATAC-seq, bulk tissue RNA-seq and CEF scRNA-seq, analysis of burst kinetics, and analysis of FACS data, prepared figures, and wrote the manuscript. SW performed ATAC-seq and ChIP-seq analyses. MHJ generated Smart-seq3 libraries. AK and JR performed Ribo-seq. JZ and XX performed analysis of tissue RNA-seq data. IP performed footprinting analysis. IC performed karyotyping. CC generated tissue RNA-seq libraries. DW conceived the study, planned and conducted chicken crossing, and performed tissue dissection. BR conceived and supervised the study, secured funding, derived the CEF lines, analysed data, prepared figures, and wrote the manuscript.

## Funding

## Competing interests

The authors declare no competing interests.
