## [Transparent Peer Review file · Nature Communications]

Multi-layered dosage compensation of the avian Z chromosome by increased transcriptional burst frequency and elevated translational rates

Corresponding Author: Professor Björn Reinius

Version 0:

Reviewer comments:

Reviewer #1

(Remarks to the Author)

In this work, Reinius et al. has adopted multi-omics methods to study an important question that how birds cope with imbalanced expression dosage between sex in the absence of global dosage compensation mechanisms on their sex chromosomes. Since female birds have only one copy of Z chromosomes, and males have two, there should be natural selection for upregulating the female Z-linked genes or downregulating the male Z-linked genes. In this work, the authors compared cell-lines of ZZ, ZW and a rare intersex ZZW genotype for their transcription and translation levels, and found that an interesting increased transcriptional burst of female Z-linked genes. This is similar to what has been found in the male mammals which upregulates their X chromosomes, and suggest convergent evolution of transcriptional regulation mechanisms of dosage compensation. In addition, the author also found from the ribosome profiling data and mass spectrometry data that female and male Z linked genes show more similar translation levels than the transcription levels. The latter point has already been reported by Lister et al. 2024 PNAS and this work confirmed the published work. The interesting new result is the increased transcriptional burst on the female Z. Overall I find the work well written and the results support their conclusions, but the new insights gained from the work are restricted to the transcriptional burst part, but this part lacks mechanistic explanations; in addition, how the transcriptional burst is associated with translational data is also unclear. I detail my comments below:

1. The authors find female Z chromosome show increased transcriptional burst frequency but not size, this is a very important pattern. However, it remains unclear why male Z do not show a burst, or more specifically, what are the molecular machinery target the female Z but not male Z to realize this female specific upregulation. Although it may sound some work that can only be addressed in another project, but with the burst result alone, the novelty derived from the work is limited. The author mentioned at line 122 that the E-box transcription factors are preferentially enriched on the female Z compared to male or the intersex. Are these E-box transcription factors biasedly expressed in females? And if so, what caused such female-biased expression pattern that may be related to the female specific transcription burst on the Z, i.e., dosage compensation. Are the binding motifs, if any, enriched on the Z relative to the autosomes, so that specific targeting onto the Z, but not autosomes can be ensured?
2. What do we know about the mechanisms of transcription burst in male mammals, regarding the targeting mechanism to the X, and male-specificity? Since the burst is the major new result, I suggest the authors introduce more about the mammalian result in the introduction.
3. I suggest the author to cite the results of Lister et al. and compare between the results, at least for the mass spectrometry. Since both studies used the same technique, although different biological samples (cell lines vs. tissues, in vitro vs. in vivo). Are the gene number with detected translation comparable between the two studies? More importantly, are the male vs. female ratio on the Z-linked genes' translation level similar between the two studies?
4. I find the rare intersex (ZZW) cell line provides very limited new insights into the problem of dosage compensation. At line 103, the authors mentioned "this unexpected individual allow us...uncouple potential effects of W on dosage compensation" There are neither previous hypotheses nor experimental results, as far as I know, support that W would impact dosage compensation. Similarly, there have been no results suggesting Y chromosome would impact dosage compensation. Because W or Y chromosomes of chicken or mammals have very few functional genes with no previous results suggesting would impact the expression level of Z or X. Could the authors please elaborate on this point? And since ZZW is not a wild type, adding this cell line just add complexity to the dosage compensation question in birds that we do not know too much about.

5. Other minor issues:

- 1) Line 36, Muller's ratchet is not the only process proposed to account for degeneration of Y chromosome, please refer to the review Bachtrög 2006.
- 2) Line 42, it needs a citation here.
- 3) Line 86, how did the authors correct for the mapping bias when estimating the allelic specific expression?
- 4) Line 104, what does it mean dosage compensated in the ZZW genotype for the Z-linked genes? As in ZZW, autosome and Z chromosomes have the same copy number. Do the authors mean down-regulation of Z linked genes? I think the so-called flexibility just means the buffering mechanism.
- 5) Line 118, if chromatin accessibility is not increased on the Z chromosome, it indicates there is no evidence for transcription upregulation from at least the ATAC-seq data. Why would that suggest 'Z upregulation is primarily controlled at transcriptional level'?
- 6) Line 120, if Fox and GATA TF show equal bindings between sexes, and between autosomes and X, why there is so-called preferential binding?
- 7) Line 136, this paragraph seems irrelevant to the major results

Reviewer #2

(Remarks to the Author)

This study uses an elegant combination of methods to uncover new insights on the mechanisms of avian Z-chromosome dosage compensation. My main suggestions for improvements concern the clarity of the presentation of the results:

1. Line 100. It is said that the unexpected triploid intersex individual (ZZW: AAA) allowed the authors to uncouple potential effects of W on Z dosage compensation. It was found that dosage compensation in the ZZW intersex individual was not mediated through Z-upregulation but by buffering of autosomal expression. However, it is unclear how this finding contributes to a better understanding of Z-dosage compensation. Does it tell us anything about the possible effects of W on dosage compensation? The buffering of autosomal expression may rather be related to maintained function of this individual despite the triploid stage (AAA). I suggest a clarification here.
2. Line 122-125 says "interestingly, E-box (enhancer box) transcription factors were enriched on female and intersex Z chromosomes compared to the male Z whereas this difference was not observed for autosomes". Suggestion: clarify why this is interesting.
3. Line 130 says "The intersex line"... suggestion: change to "intersex individual" or "intersex cell line".
4. Discussion line 237-244. Here it is first concluded that Z-upregulation is driven by increased burst frequency, but not size, resembling the kinetic mode of the evolutionary distinct X-upregulation observed in mouse. Then it is stated that the core promoter elements controlling transcriptional burst size remain the same. What is then the relevance of the latter molecular mechanism in this context?
5. Discussion lines 256-257 conclude that Z-upregulation results from female cells carrying only one Z allele, not the presence of W. Lines 263-265 says "Intriguingly, similarly Z-enriched ribosomal density was observed in ZZW intersex cells suggesting the involvement of W-linked factors". Comment: I agree that the apparently nuanced role of the W chromosome (i.e. not promoting Z-upregulation but possibly promoting increased translational efficiency of Z-transcripts in females through higher ribosomal density) is interesting. Is there any idea how this mechanism could be associated with the W chromosome?
6. Discussion lines 272-275, conclude that there is an unexpected similarity to mammalian dosage compensation when all regulatory layers are coherently taken into consideration. Comment: In what way is Z chromosome dosage compensation unexpectedly similar to X chromosome dosage compensation? In terms of overall compensation or in terms of the interplay between the underlying mechanisms?
7. Abstract. Clarify which of the revealed mechanisms that are "remarkable conserved" and which mechanisms that are "new". In what way is this study highlighting the importance of gene-dosage balance across diverse species?

Version 1:

Reviewer comments:

Reviewer #1

(Remarks to the Author)

This is a revised version of a work characterizing the mechanism of dosage compensation, if any, of birds, using multi-omics approaches on chicken cell lines. I don't think the authors have fully addressed my previous questions and I detailed my comments below. The limitation of novelty and the necessity and value of using a triploid cell line are my previous major concerns. Instead of answering directly to the questions, the authors reiterated how much new data they have generated.

This does not justify the novelty but reflects lack of clear hypothesis and corresponding experimental design. Again I acknowledge the discovery of transcriptional bursting in chicken, in parallel to that of mammals, which I think is important for such a convergent pattern. However, without knowing the clear targeting mechanism on the X in mammals, or on the Z in chicken, based on what the authors have answered in this revision, it is rather descriptive. In this revision, the authors also found a contradiction compared their results to Lester et al. PNAS, which confounds the conclusion of this work. Finally I think the title "...in birds" is misleading and an overstatement, given that only cell lines of chicken were studied here.

1. About the novelty. A general issue is that the authors tried to emphasize what they have done, instead of why this is necessary or important in their response of justifying the novelty. If no clear hypothesis or aim is laid out for massive work, it reflects a poor design of experiment.

1) Indeed, the authors generated bulk RNA-seq, scRNA-seq, ATAC-seq, ChIP-seq, Ribo-seq and MS data in this work. But these techniques are as regular as PCR nowadays for any molecular genomics labs, and one cannot appreciate the difficulty of collecting these data from chicken.

2) Why is it important to study pure-bred and cross-bred chicken, if the question of the project is to study whether there exists and what is the mechanism of dosage compensation in birds? What is the explanation if there is, or not difference between pure-bred vs. cross-bred chicken in the findings of dosage compensation?

3) This is related to the question that I raised in the last round of revision. There is no indication at all that Y chromosome, or W chromosome would impact the dosage compensation from any species. So it only reflects a poor design of experiment and lack of hypothesis, when the authors decided to study a ZZW cell line and examine whether there is allelic expression or not. The authors mentioned below ZZW is from a wild type chicken, but how rare is that? This is like saying to study an aneuploidy cell line so that one can understand the mechanism of gene regulation in a normal individual without aneuploidy, which does not make sense.

4) This is a novel point, as I mentioned in the previous comment and also here above.

5) The same as 4)

6) Again the authors were describing what they did without elucidating why they did it. In addition, from ATAC-seq and ChIP-seq, the authors found no difference between male and female Z chromosome, which provides no evidence for dosage compensation from these two datasets at all! How generating a new data without generating a new and relevant conclusion can justify the novelty of the work?

7) So there is a contradiction between the Lester et al. vs. this work in their conclusion. Basically Lester et al. from in vivo situation found evidence for dosage compensation, but the authors here did not, as the MS data here showed there is a sex-bias in protein level. (see more of my comments below)

8) I don't think Ribo-seq on chicken is 'pioneering' at all: this was done in chicken five years ago (Zhong-Yi et al 2020 Nature)

9) See point 3) above.

2. I asked in the last round regarding the targeting mechanism of Z that leads to transcription burst preferentially occurred in female but not male chicken. The authors mentioned E-box transcription factor as a potential responsible mechanism. However, during this revision, the authors indicated that this transcription factor is not female-biased from transcriptomic data. The binding motifs of E-box TF are also not enriched on the Z vs. autosomes. Then how would this explain the role of E-box transcription factor, if any, in targeting specifically the female Z chromosome? The only suggestive evidence seems to be 'a preferential enrichment of E-box-binding transcription factor' on the female Z relative to male Z. But I did not find in the method (it only showed the motif search) how this conclusion was reached, did the author perform ChIP targeting the E-box-binding TF in male and female cell lines?

3. The authors answered my second question about the targeting mechanism to the X by citing their published work, except for how exactly the X chromosome is targeted for transcription bursting. I wonder what transcription factors (supposed to be male biased or specific) and binding motifs are involved in mammalian X transcription bursting?

4. A key difference between this work vs. Lister et al. 2024 is that based on MS data, the authors find in this work DO NOT support dosage compensation in the translation level, while Lister et al. does. The authors attributed the difference to more sensitive MS results from this work, but then the conclusion is confusing. Does this mean dosage compensation in translation only occurs for highly translated protein? In addition, the Lister et al. 2024 used in vivo tissues, while the authors in this work did MS on cell lines. This is also an important confounding factors, and I generally tend to think an in vivo tissue likely reflect better the dosage compensation than the cell lines.

5. For the answers to my previous question 4, the authors justified using a ZZW cell line because 'intersex sample allowed us to evaluate whether the W chromosome contributes to the Z-linked dosage compensation'. But why there is a need to evaluate this? As I previously asked, there was no evidence or reasoning to think that the Y or W would impact dosage compensation given both chromosomes have few functional genes. Could the authors elaborate by what mechanisms the Y/W would impact dosage compensation? And why it is important to examine whether Y/W would have such an impact, if any. Also, the authors also responded "increased Z chromosome burst frequency in females comes from having only one Z copy". I did not understand, what is the alternative given female only have one Z, and if there is a Z-linked burst, it must come from the only Z.

Reviewer #2

(Remarks to the Author)

I think that the authors have made an overall good job in revising their manuscript. The changes made have increased the clarity of the study. I only have one remaining concern.

1. This is about the interpretations of the expression patterns found in the triploid intersex individual (ZZW:AAA). The reduced expression of the autosomal genes is interpreted as a mechanism for dosage compensation of the Z chromosome. However, a reduced expression of the autosomal genes could also be a mechanism for buffering autosomal expression per se to maintained function and viability of triploid individuals. In their answer the authors say "We do agree that buffering of autosomal expression may be related to maintained function and viability of triploid individuals". However, in the changed text in the manuscript the reduced expression of autosomal genes is still only viewed as a mechanism of Z-chromosome upregulation. Thus, some further clarifications of why this is the only or most likely interpretation is needed.

Version 2:

Reviewer comments:

Reviewer #2

(Remarks to the Author)

I am happy with the authors' response to my last issue raised.

Remaining main objections from reviewer 1 are that the authors lack clear hypothesis and highlight their methods rather than pinpointing what their novel findings are (i.e. that they describe what they have done rather than why). I think that the authors in general have managed to solve these issues in the revision of the ms. However, some parts of the Discussion section would benefit from further increased clarity.

1.Lines 296-302 in the discussion. This paragraph starts with "Chromatin accessibility have been expected to at least partly underlay X-upregulation in mouse..." the authors then explains what they find: "... we observed no differences in accessibility or chromatin state between female and male z-chromosomes" and conclude that this hints to an "additional level of similarity between mammalian and avian dosage compensation mechanisms."

Suggestion: This paragraph would benefit from clarification. Is the main similarity between birds and mammals that chromatin accessibility is not a major mechanism of X/Z-upregulation? (even if some early studies suggested so in mouse).

2. Lines 303-314 in the discussion nicely describes how the ZZW chicken was used (and why) but lack references to previous studies making it difficult for the reader to judge how novel the finding is. Even if this indeed is explained in other parts of the ms, it needs to be re-stated here to make the discussion stand by itself.

3. Minor suggestion 137-139: To investigate whether the chromatin landscape of the Z chromosome contributes to the regulation of Z-upregulation..."

Suggestion: Change to "To investigate whether the chromatin landscape of the Z chromosome contributes to Z-upregulation..."

RESPONSE TO REVIEWERS' COMMENTS

Reviewer #1 (Remarks to the Author)

In this work, Reinius et al. has adopted multi-omics methods to study an important question that how birds cope with imbalanced expression dosage between sex in the absence of global dosage compensation mechanisms on their sex chromosomes. Since female birds have only one copy of Z chromosomes, and males have two, there should be natural selection for upregulating the female Z-linked genes or downregulating the male Z-linked genes. In this work, the authors compared cell-lines of ZZ, ZW and a rare intersex ZZW genotype for their transcription and translation levels, and found that an interesting increased transcriptional burst of female Z-linked genes. This is similar to what has been found in the male mammals which upregulates their X chromosomes, and suggest convergent evolution of transcriptional regulation mechanisms of dosage compensation. In addition, the author also found from the ribosome profiling data and mass spectrometry data that female and male Z linked genes show more similar translation levels than the transcription levels. The latter point has already been reported by Lister et al. 2024 PNAS and this work confirmed the published work. The interesting new result is the increased transcriptional burst on the female Z. Overall I find the work well written and the results support their conclusions, but the new insights gained from the work are restricted to the transcriptional burst part, but this part lacks mechanistic explanations; in addition, how the transcriptional burst is associated with translational data is also unclear. I detail my comments below:

We sincerely thank Reviewer 1 for their thoughtful comments and for recognizing the significance of our work in addressing important questions. We also appreciate the positive feedback on the clarity of our writing and the strength of our conclusions, which you find well-supported by our results. Your encouraging words are appreciated and exemplify the role of a constructive and supportive reviewer.

However, we respectfully disagree with the reviewer's statement that the new insights from our work are limited to the transcriptional burst aspect. Our paper presents multiple other novel findings on avian dosage compensation (see list below). That said, we believe the first-ever characterization of bursting kinetics in birds is, even on its own, a significant breakthrough. We acknowledge that some of the novel aspects may not have been as explicitly apparent, partly due to the manuscript's compact formatting. This stems from its initial preparation as a condensed Letter, being transferred from another journal. In the revised version, now formatted according to Nature Communications guidelines with subheadings for key sections, the presentation is significantly clearer. We hope this revision highlights the broad scope and comprehensive nature of our study.

To unequivocally highlight the novelty of our study, we outline its key contributions below:

- The most comprehensive unified original dataset of avian dosage compensation spanning: bulk RNA-seq, single-cell RNA-seq, ATAC-seq, ChIP-seq, Ribo-seq, proteomics (MS2 and MS3), and chromosome copy-number control by DNA-seq and karyotyping.
- First use of cross-bred chickens to dissect dosage compensation in birds at allelic resolution, demonstrating strong evidence of Z-chromosome-wide upregulation affecting the single female Z chromosome across both pure-bred and cross-bred tissue samples, and in vivo tissues and in vitro cells (CEF primary cell lines).
- Use of allele-resolved single-cell RNA-sequencing to dissect Z-chromosome dosage compensation allele-by-allele at cellular resolution. These data also provide the first examination of dynamic random monoallelic expression in birds.
- We provide the first ever examination of transcriptional burst kinetics in birds, demonstrating that the core promoter elements controlling transcription burst size are shared between mammals and birds, despite 310 million years of evolution between them.
- We provide strong evidence demonstrating that female-specific Z-chromosome upregulation is driven by increased transcriptional burst frequency, mechanistically resembling the independently evolved mammalian X chromosome upregulation, pointing to convergent evolution of dosage compensating mechanisms in mammals and birds.
- We provide allele-level characterization of chromatin accessibility (ATAC-seq) and chromatin landscapes (quantitative ChIP-seq) in the context of Z-chromosome dosage compensation.
- Through extensive proteomic analysis including two rounds of fractionation (MS2 and MS3), we suggest a near-complete (~80%) Z-chromosome dosage compensation on the proteomic level. Importantly, the one previous study performing proteomic analyses of dosage compensation in birds (Lester et al), mentioned by the reviewer, reported a complete lack of any sex-bias at the protein level – likely as a result of being underpowered to detect the 1.2-fold difference revealed by our mass spectroscopy data (See response to comment X below)
- Pioneering ribosomal profiling of dosage compensation in birds, we show for the first time in birds, that Z chromosome dosage compensation is partially driven by increased translation efficiency of Z-chromosome transcripts in females.

- Through the inclusion of a rare but naturally occurring ZZW triploid sample, we uncouple the effects of Z-chromosome copy number and the presence of a W chromosome.
- Overall, our study offers multiple novel insights into the regulation of avian dosage compensation. When integrating all regulatory layers, an unexpected similarity between avian and mammalian dosage compensation emerges.

We hope that the reviewer agrees that these data and insights constitute a significant contribution to the field.

Based on the reviewers' suggestions, we have restructured the manuscript to enhance clarity. All text changes since the previous version are highlighted in yellow. Additionally, we have reorganized some of the main figures for better clarity, expanding from three to four main figures and incorporating selected supplementary panels into the main figures. We hope the reviewer finds the revised structure improved.

1. The authors find female Z chromosome show increased transcriptional burst frequency but not size, this is a very important pattern. However, it remains unclear why male Z do not show a burst, or more specifically, what are the molecular machinery target the female Z but not male Z to realize this female specific upregulation. Although it may sounds some work that can only be addressed in another project, but with the burst result alone, the novelty derived from the work is limited. The author mentioned at line 122 that the E-box transcription factors are preferentially enriched on the female Z compared to male or the intersex. Are these E-box transcription factors biasedly expressed in females? And if so, what caused such female-biased expression pattern that may be related to the female specific transcription burst on the Z, i.e., dosage compensation. Are the binding motifs, if any, enriched on the Z relative to the autosomes, so that specific targeting onto the Z, but not autosomes can be ensured?

We thank the reviewer for raising this point. However, it is incorrect to suggest that the novelty of our work is limited to the transcriptional bursting aspect—please see our previous response and the list of novel features above. We furthermore do agree that dissecting the molecular machinery of the bursting regulation of Z upregulation is another project in itself, or more likely several projects.

Regarding E-box transcription factors: We indeed observed a preferential enrichment of E-box-binding transcription factors on both female and intersex Z chromosomes compared to the male Z. Using our transcriptomic data, we did not detect female-biased or intersex-biased expression of E-box transcription factors. While outside of the scope of our current work, one could speculate that E-box transcription factors may be involved in sex-specific Z-chromosome gene regulation. Of note, this enrichment of E-box-binding transcription factors was maintained even when the MHM region was excluded from the analysis, suggesting that this enrichment is not MHM-specific. Our analysis showed that the Z chromosome is not enriched in E-box motifs compared to autosomes of similar size ($P = 0.16$, one-sample Wilcoxon test, observed/expected enrichment ratio = 0.95 compared to chromosomes of similar size) again pointing to a potential involvement of E-box-binding motifs in female- and intersex-specific gene expression rather than Z-chromosome upregulation. We have included a new Supplementary table (Supplementary table 6) containing the chromosome length in bases, number of E-box motifs per chromosome and number of E-box motifs per non-N-masked base.

We have added the following text to our **Results** section to better convey our findings regarding E-box motifs:

*“Through a global analysis of transcription factor footprinting using ATAC-seq data, we examined differential transcription factor binding by performing pairwise comparisons of the Z chromosome and autosomes between the sexes. We observed preferential enrichment of the FOX transcription factor family on the Z chromosome and autosomes in males, while the GATA transcription factor family was enriched on the Z chromosome and autosomes in females (**Supplementary Fig 4d-e, Supplementary table 4**). E-box-binding transcription factors were enriched on female and intersex Z chromosomes compared to the male Z, whereas this difference was not observed for autosomes (**Supplementary Fig. 4d-e, Supplementary table 4**). Notably, E-box motif sequences were not enriched on the Z chromosome compared to autosomes (**Supplementary table 5**) potentially, hinting to a sex-specific role in Z-gene regulation which could be explored in future studies.”*

2. What do we know about the mechanisms of transcription burst in male mammals, regarding the targeting mechanism to the X, and male-specificity? Since the burst is the major new result, I suggest the authors introduce more about the mammalian result in the introduction.

We and colleagues have previously characterized the genomic encoding of transcriptional burst kinetics in mammals [Larsson *et al.*, 2019 *Nature*]. Subsequently, we described increased transcriptional burst frequency associated with X-chromosome upregulation in mammalian cells [Larsson *et al.*, 2019 *Nature Structural and Molecular Biology* & Lentini *et al.*, 2022 *Nat Comms*], demonstrating increased bursting for the single active X allele in both XY males and XX females. Given these patterns, we therefore think that it is extremely interesting that we can now demonstrate burst-frequency-driven also of an avian Z-chromosome, which is evolutionarily unrelated to the mammalian X. This highlights that this mechanism has evolved convergently between ZW and XY systems. We agree with the reviewer that additional background information on bursting will help the reader, and have therefore expanded the text with the following sentences to better introduce the concepts of transcriptional burst kinetics:

In the Introduction:

“...Finally, eukaryotic transcription occurs in stochastic bursts of RNA synthesis from each alleles^{16–20}. However, despite its crucial role in understanding expression regulation, the kinetics of transcriptional bursting and its genomic encoding remain entirely unexplored in birds...”

In the Results:

“...At the single-cell level, eukaryotic transcription is inherently stochastic and occurs in short bursts of activity from the individual alleles^{41,42}. Transcriptional kinetics can be represented by burst frequency (rate of transcription pulses) and burst size (average number of molecules produced per burst) within the two-state telegraphic model of transcription. We and colleagues, previously characterized the genomic encoding of transcriptional burst kinetics in mouse from allelic scRNA-seq data¹⁶. Moreover, we showed that X-upregulation in mouse is primarily driven by increased transcriptional burst frequency, and not increased burst size^{28,29}. Here, we sought to characterize transcriptional kinetics in birds for the first time. To this end, we inferred parameters of transcriptional bursting (burst frequency: k_{on} and burst size: k_{off}) for each allele using our Xpress-seq data (see Methods and Supplementary Fig. 9a-b)...”

Furthermore, we now guide the reader a bit more in the interpretation of our Results, e.g:

“...Importantly, our data suggest that the control principles of burst kinetics are universally shared across mammals and birds, with the genomic encoding of transcriptional bursting being highly conserved among vertebrate species...”

We understand that several concepts in transcriptional burst kinetics may be unfamiliar to some readers, as global characterization of bursting (using allele-specific single-cell RNA-seq) was only recently made possible. We hope these sentences will help provide context for our results. Thank you for highlighting the need for clarification.

3. I suggest the author to cite the results of Lister et al. and compare between the results, at least for the mass spectrometry. Since both studies used the same technique, although different biological samples (cell lines vs. tissues, in vitro vs. in vivo). Are the gene number with detected translation comparable between the two studies? More importantly, are the male vs. female ratio on the Z-linked genes' translation level similar between the two studies?

We thank the reviewer for their comment, and we agree that this is merited. We indeed already cited Lister et al. 2024 in the previous version on the manuscript, but an explicit comparison was lacking. The findings from Lister et al. 2024 overall agree with our results regarding RNA-seq and proteomic measurements. Important to note, however, is that the Lister study was limited to bulk RNA-seq and proteomics (MS/MS), while our study provides multiple other layers of information (see previous points above).

While the methodologies of protein detection by mass spectroscopy are similar in Lister et al and our study, the results differ as we indeed detect a small but significant male bias remaining at the protein level (male-to-female expression ratio: 1.2-fold, $P < 8.6 \times 10^{-7}$). This discrepancy in results may stem from differences in tissues analysed (liver and heart versus fibroblasts) but more likely from increased statistical power of our experiments, capturing more than twice the number of proteins and maintaining a strong correlation between RNA and protein ratios (Please see numbers and plots further down in this response). We do further note that the Lister et al. study lacks any evidence for how dosage compensation is achieved in chicken, where we provide evidence from multiple regulatory layers. Importantly, we performed Ribo-seq (not done in Lister et al) which demonstrated increased translational rates (increased ribosomal occupancy) on Z-linked transcript in females compared to males. We agree that further comparison with the Lister et al data is useful and have now included the **new Supplementary Figure 14**, which summarizes the results of our re-analysis.

Are the gene number with detected translation comparable between the two studies?

Lister and colleagues report the detection of 721-836 autosomal and 43-48 Z-linked proteins in chicken liver and heart. Our most sensitive round of fractionation (MS3) resulted in the detection of 2031 autosomal and 91 Z-linked proteins across samples, more than doubling protein detection in our dataset compared to Lister et al. and thereby increasing sensitivity.

Regarding the number of genes detected on both the RNA and protein level: Lister et al report the detection of 578-668 autosomal and 32-37 Z-linked genes in chicken tissues, detected on both the RNA (RNA-seq) and protein (MS/MS) levels. We detected 1167 autosomal and 53 Z-linked genes using MS2, and 1934 autosomal and 90 Z-linked genes using MS3.

More importantly, are the male vs. female ratio on the Z-linked genes' translation level similar between the two studies?

On the transcriptome level Male:Female ratios are very comparable between our findings and those of Lister et al. (our data: CEF Z-chr M:F ratio (RNA) = 1.54, Lister: chicken heart Z-chr M:F 1.43, chicken liver Z-chr M:F=1.35, **Reviewer fig. 1a, RNA panels**). On the translational level, we observed increased ribosomal density for female Z-linked transcripts compared to male Z-linked transcripts and autosomal transcripts, using ribosome profiling (Ribo-seq). Importantly, no ribosomal density measurements were made in Lister et al. On the proteomic level, Lister et al. report that "*median M:F ratios of the proteins encoded by Z-borne genes were near 1:1 in both chicken tissues*". While we find comparable protein-level M:F ratios for Z-borne genes (**Lister et al.** chicken heart Z-chr M:F = 1.1, chicken liver Z-chr M:F = 0.93; **our data:** MS2 data: Z-chr M:F = 1.17, MS3 data: Z-chr M:F = 1.13, **Reviewer figure 1a-b**), increased M:F ratios for Z-linked genes were statistically significant when compared to autosomal M:F ratios (MS2: $P = 8.3 \times 10^{-7}$, MWU test, MS3: $P = 9.198 \times 10^{-7}$, MWU test, **Reviewer figure 1b, Protein-MS2, Protein-MS3 panels**), showing that there is still a difference in Z-linked protein levels between males and females. This is an important distinction between ours and Lister's data. Additionally, we observed a higher degree of correlation between RNA-level Male:Female ratios and protein-level Male:Female ratios for our datasets compared to Lister et al. (**Reviewer Figure 1c-d**), pointing to a higher degree of overall agreement between the measurements in our data.

Reviewer Figure 1. Re-analysis of Lister et al. 2024 RNA-seq and proteomic datasets of chicken liver and chicken heart tissues.

a, Boxplots of transcriptome (RNA-seq) and proteome (mass spectrometry) - level Male:Female gene expression ratios for autosomes and Z chromosome obtained from Lister et al.2024. Dashed lines indicate median. (Heart: autosome-RNA = 0.98, Z-RNA = 1.43, protein-autosome = 1.03, protein-Z = 1.10. Liver: autosome-RNA = 0.99, Z-RNA = 1.35, autosome-protein = 0.95, Z-protein = 0.93). Mann-Whitney-U test used for significance testing.

b, Boxplots of transcriptome- (RNA-seq) and proteome- (MS2 and MS3) level Male:Female gene expression ratios for autosomes and Z chromosome obtained from . Dashed lines indicate median (autosome-RNA = 0.96, Z-RNA = 1.45, autosome-MS2 = 0.98, Z-MS2 = 1.14, autosome-MS3 = 0.98, Z-MS3 = 1.13). Number of genes detected in both RNA-seq and proteomics shown at the bottom left of the respective boxplots for autosomes (red) and the Z chromosome (in blue). Mann-Whitney U test used for significance testing.

c, Scatterplots of Male:Female ratios of RNA-level (y-axis) and protein-level (x-axis) gene expression for the Z-

chromosome (purple, left panels) and autosomes (green, right panels) for chicken heart (top) and chicken liver (bottom) tissues using data from Lister et al. 2024. Number of genes detected in both the RNA-seq and mass spectrometry experiments, Pearson's rho and p-values shown at the bottom right corner of each scatterplot n=3 for male and female chicken heart, n=2 for male and female chicken liver.

d, Scatterplots of Male:Female ratios of RNA-level (y-axis) and protein-level (x-axis) gene expression for the Z-chromosome (purple, left panels) and autosomes (green, right panels) for chicken embryonic fibroblast (CEFs) using MS2 (Second-stage Mass spectrometry) or MS3 (Third-stage Mass Spectrometry). Number of genes detected in both RNA-seq and MS2 (top) or RNA-seq and MS3 (bottom), Pearson's rho and p-values shown at the bottom right corner of each scatterplot. n=2 male and n=2 female primary chicken embryonic fibroblast lines, in addition to 3 independent technical replicates for each experiment.

4. I find the rare intersex (ZZW) cell line provides very limited new insights into the problem of dosage compensation. At line 103, the authors mentioned “this unexpected individual allow us...uncouple potential effects of W on dosage compensation” There are neither previous hypotheses nor experimental results, as far as I know, support that W would impact dosage compensation. Similarly, there have been no results suggesting Y chromosome would impact dosage compensation. Because W or Y chromosomes of chicken or mammals have very few functional genes with no previous results suggesting would impact the expression level of Z or X. Could the authors please elaborate on this point? And since ZZW is not a wild type, adding this cell line just add complexity to the dosage compensation question in birds that we do not know too much about.

We recognize that some clarifications are needed regarding the importance of the ZZW individual in our study and have therefore added further explanation. We would also like to emphasize that the triploid is, in fact, a "wild type" natural phenomenon in birds—though rare—which makes the inclusion of this intersex sample even more significant.

The addition of the triploid intersex individual (ZZW) presented us with the unique opportunity to characterize sex chromosome dosage compensation in a system carrying both the normally “female-specific” W chromosome and the normally “male-specific” dual dose (WW) of chromosome Z. Therefore, intersex sample allowed us to evaluate whether the W chromosome contributes to Z-linked dosage compensation, or whether Z-upregulation is independent of the presence of the W chromosome. Through this approach, we were able to demonstrate that in the ZZW intersex cells, Z-upregulation did not occur in the same way as in the ZW females, namely there was a lack of transcriptional Z hyperactivation in ZZW cells. Crucially, this lets us draw the conclusion that increased Z-chromosome burst frequency in females comes from having only one Z copy, similarly to the situation for chromosome X in mammals (and not from any W-encoded factor). On the other hand, we observed slightly increased ribosomal density on Z transcripts in the ZZW system, potentially suggesting some involvement of W in this action (which would not be impossible since W,

albeit gene-poor, contain ribosomal genes). In addition, we investigated the expression of the MHM region, which is normally specifically expressed in females, in ZZW samples, revealing expression from both Z-copies together with open chromatin in the ZZW genotype. Finally, we investigated the expression of miR-2954 (involved in reducing Z-transcript levels in males) in ZZW for the first time and found it to be expressed at intermediate level.

We have tried to improve the texts surrounding the ZZW samples throughout the manuscript. Although we agree that many questions of dosage compensation (e.g. the nature of autosomal dampening) need further investigation in future studies, we do hope that it is now more apparent why a characterization of a ZZW sample is both significant and exciting.

5. Other minor issues:

1) Line 36, Muller's ratchet is not the only process proposed to account for degeneration of Y chromosome, please refer to the review Bachtrog 2006.

We agree. Thanks for spotting. We added reference to Bachtrog 2006, and we simplified this passage. The sentence now reads:

Although the systems evolved independently, both originated from an autosomal chromosome pair and experienced degeneration of the non-recombining sex chromosome^{2,3}, which defines the heterogametic sex.

2) Line 42, it needs a citation here.

We agree and have now included the following references in this passage (now around Line 48):

Nguyen, D. K. & Disteche, C. M. Dosage compensation of the active X chromosome in mammals. Nat. Genet. 38, 47–53 (2006).

Deng, X. et al. Evidence for compensatory upregulation of expressed X-linked genes in mammals, Caenorhabditis elegans and Drosophila melanogaster. Nat. Genet. 43, 1179–1185 (2011).

Cecalev, D., Viçoso, B. & Galupa, R. Compensation of gene dosage on the mammalian X. Development 151, dev202891 (2024).

3) Line 86, how did the authors correct for the mapping bias when estimating the allelic specific expression?

A full methodological description of this process is provided under “Materials and methods”, section “*Variant calling and allelic quantification*”. In essence, we N-masked strain-specific variants to avoid mapping bias to any specific strain. We have updated the mentioned passage in the Results section stating that N-masking was used, which is important for limiting mapping bias. Thanks for highlighting. Note also that all computational code used in our analyses are furthermore available at Github (stated under Code availability).

4) Line 104, what does it mean dosage compensated in the ZZW genotype for the Z-linked genes? As in ZZW, autosome and Z chromosomes have the same copy number. Do the authors mean down-regulation of Z linked genes? I think the so-called flexibility just means the buffering mechanism.

The ZZW individual presented in this study has a triploid genotype, including 3 copies of each autosome and two copies of the Z chromosome (as well as a single copy of the W chromosome). In the absence of dosage compensation, we would expect a 30% higher expression in autosomes compared to the Z chromosomes. However, we observe that the autosomes fail to reach the expected 30% increase in transcriptional output, suggesting that the stoichiometric balance between the autosomes and the Z chromosomes in these individuals does not rely on Z-chromosome upregulation but rather on the downregulation of autosomes. We would like to highlight that we do not observe any downregulation of Z-linked genes in ZZW samples. We agree that the term "flexibility" was inappropriate, so we have removed it and revised the text for greater clarity. The passage now reads:

*Expression relative to autosomes revealed that Z-linked expression was partially compensated in both ZW and ZZW genotypes (**Supplementary Fig. 3a**). Interestingly, whereas RNA-seq of CEF cell lines reconfirmed Z-upregulation in ZW females on par with in vivo tissues (mean fold-change 1.54 in CEFs, **Fig. 2b**), dosage compensation in the ZZW intersex individual was not mediated through Z-upregulation but through buffering of autosomal expression (**Supplementary Fig. 3b**), i.e. less than linearly additive, as has been observed for other species³³.*

5) Line 118, if chromatin accessibility is not increased on the Z chromosome, it indicates

there is no evidence for transcription upregulation from at least the ATAC-seq data. Why would that suggest 'Z upregulation is primarily controlled at transactional level'?

This now corresponds to Line 137 in the updated text. While chromatin accessibility is highly dynamic throughout development, it is less dynamic in stable cell states -and importantly- transcription can be fine-tuned through modes not involving change in chromatin accessibility (instead e.g. frequency of enhancer interactions in line with a burst-frequency-driven regulation). We thus reason that if there are no differences in chromatin accessibility on individual Z chromosomes, there is not an inherently permissive chromatin landscape that would facilitate increased expression. Additionally, we observed that on the transcriptional level, the female Z is upregulated to achieve up to 50% higher output, mediated through increased transcriptional burst frequency. Although we find that increased translation efficiency of Z-linked transcripts is higher in females, the degree of dosage compensation achieved by Z-chromosome transcriptional upregulation compared to all other modes examined in this work suggest that the main mode of regulation is at the transcriptional level. We have updated the text to read as follows:

Results section: “ *Similar to our previous findings of X-upregulation in mice³⁰, chromatin accessibility was not increased on the upregulated Z allele despite pronounced Z-upregulation on the transcriptional level (Fig. 2c and Supplementary Fig. 4a-c), suggesting limited involvement of chromatin accessibility in regulating Z-upregulation.*”

6) Line 120, if Fox and GATA TF show equal bindings between sexes, and between autosomes and X, why there is so-called preferential binding?

In this instance, we use the term “preferential binding” to refer to the difference in enrichment between the male samples and the female samples, rather than between the autosomes and the Z chromosome.

To better convey this, the respective text now reads as follows:

“*Through a global analysis of transcription factor footprinting using ATAC-seq data, we examined differential transcription factor binding by performing pairwise comparisons of the Z chromosome and autosomes between the sexes. We observed preferential enrichment of the FOX transcription factor family on the Z chromosome and autosomes in males, while the GATA transcription factor family was enriched on the Z chromosome and autosomes in females (Supplementary Fig 4d-e, Supplementary table 4). E-box-binding transcription factors were enriched on female and intersex Z chromosomes compared to the male Z,*

whereas this difference was not observed for autosomes (Supplementary Fig. 4d-e, Supplementary table 4). Notably, E-box motif sequences were not enriched on the Z chromosome compared to autosomes (Supplementary table 5) potentially, hinting to a sex-specific role in Z-gene regulation which could be explored in future studies.”

7) Line 136, this paragraph seems irrelevant to the major results

This is now line 142. We thank the reviewer for their input as it allows us to provide a better explanation for this paragraph. Previous work in human and mouse reported that cell size is correlated to the number of genes expressed biallelically due to stochastic allelic transcription [Reinius et al, 2016 Nature Genetics, Reinius & Sandberg 2015 Nature Reviews Genetics]. However, this has not been investigated at all in birds. Since our work includes the first allelically-resolved characterization of transcriptional bursting in birds, we took the opportunity to answer the question whether biallelic expression is correlated to cell size as has been previously shown in mammals. Although these results are not directly relevant to the topic of sex chromosome dosage compensation, they provide another example of that fundamental principles of stochastic allelic transcription and kinetic modalities are conserved between mammals and birds. Therefore, this result provides a valuable piece of information for the research community in transcriptional burst kinetics that we wish to keep in the paper.

We would like to once again thank Reviewer 1 for their valuable comments, which we believe that we have thoroughly addressed. We hope that you now find our paper ready for publication.

Reviewer #2 (Remarks to the Author):

This study uses an elegant combination of methods to uncover new insights on the mechanisms of avian Z-chromosome dosage compensation. My main suggestions for improvements concern the clarity of the presentation of the results:

We sincerely thank the Reviewer for their thoughtful comments and for recognizing the elegance and novelty of our study. We appreciate the suggestions for improving the clarity of

our results and have made revisions accordingly. We hope the Reviewer finds the presentation in the revised version of the manuscript clearer and more refined.

Based on the reviewers' suggestions, we have restructured the manuscript to enhance clarity. All text changes since the previous version are highlighted in yellow. Additionally, we have reorganized some of the main figures for better clarity, expanding from three to four main figures and incorporating selected supplementary panels into the main figures. We hope the reviewer finds the revised structure improved.

1. Line 100. It is said that the unexpected triploid intersex individual (ZZW: AAA) allowed the authors to uncouple potential effects of W on Z dosage compensation. It was found that dosage compensation in the ZZW intersex individual was not mediated through Z-upregulation but by buffering of autosomal expression. However, it is unclear how this finding contributes to a better understanding of Z-dosage compensation. Does it tell us anything about the possible effects of W on dosage compensation? The buffering of autosomal expression may rather be related to maintained function of this individual despite the triploid stage (AAA). I suggest a clarification here.

This now corresponds to lines 119-121 in the updated text.

We agree that further clarification would improve readability here. It has been previously suggested that Z-chromosome upregulation may be controlled by a W-linked factor (Graves 2003, *Cytogenetic and Genome Research*), which may act (potentially as a transcription activator) on the lncRNAs encoded by the MHM region which is located within chrZ, and showing both chromatin accessibility and strong expression in females but not males. In turn, the lncRNAs of the MHM region may mediate chromosome-wide Z-chromosome upregulation, which could potentially limit the ability to control Z-chromosome upregulation to female chickens and would provide an explanation as to why diploid Z0 and diploid ZZW chickens are non-viable. However, in the case of the triploid ZZW, the presence of a transcriptionally active W chromosome (as shown by the expression of HINTW and SPINW) is not associated with Z-chromosome upregulation, which led us to conclude that Z-chromosome upregulation is not dependent on the presence of a W chromosome. We do agree that buffering of autosomal expression may be related to maintained function and viability of triploid individuals and have updated the text to reflect this:

In the Results section:

“Interestingly, whereas RNA-seq of CEF cell lines reconfirmed Z-upregulation in ZW females on par with in vivo tissues (mean fold-change 1.54 in CEFs, Fig. 2b), dosage compensation

in the ZZW intersex individual was not mediated through Z-upregulation but through buffering of autosomal expression (Supplementary Fig. 3b), i.e. less than linearly additive, as has been observed for other species³³, suggesting that Z-chromosome upregulation is not dependent on the presence of a W chromosome.”

In the Discussion section:

“Notably, while both female and intersex samples carry a W chromosome, we found no evidence of transcriptional Z-upregulation in the ZZW genotype. Conversely, we observed that incomplete dosage compensation in intersex cells was mediated by the reduced expression of autosomal genes, which would improve the stoichiometric balance and contribute to the viability of these individuals. Thus, Z-upregulation by increased transcriptional burst frequency in female cells results from carrying only one Z allele, and not from the presence of W. Despite the presence of two Z chromosomes in intersex, we found the MHM region to be accessible and transcriptionally active in both females and intersex, indicating that MHM expression is dependent on the presence of a W and not the number of Z chromosomes.”

2. Line 122-125 says “interestingly, E-box (enhancer box) transcription factors were enriched on female and intersex Z chromosomes compared to the male Z whereas this difference was not observed for autosomes”. Suggestion: clarify why this is interesting.

E-box transcription factors are important developmental regulators, suggesting there may be a sex-specific role for these factors. While exploring this is outside the scope of the current study, we found this finding interesting. Of note, this effect remains in female and intersex Z chromosomes even after excluding the MHM region which is enriched for E-box motifs. Furthermore, we observed that this enrichment is not due to a disproportionate enrichment of E-box motifs on the Z chromosome compared to autosomes of the same size (Supplementary table 5), which may indicate that the E-box transcription factor enrichment we observe may be involved in female- and intersex-specific gene expression rather than Z upregulation.

We have added the following text to our Results section to better convey around findings:

“Through a global analysis of transcription factor footprinting using ATAC-seq data, we examined differential transcription factor binding by performing pairwise comparisons of the Z chromosome and autosomes between the sexes. We observed preferential enrichment of the FOX transcription factor family on the Z chromosome and autosomes in males, while the GATA transcription factor family was enriched on the Z chromosome and autosomes in females (Supplementary Fig 4d-e, Supplementary table 4). E-box-binding transcription factors were enriched on female and intersex Z chromosomes compared to the male Z,

whereas this difference was not observed for autosomes (Supplementary Fig. 4d-e, Supplementary table 4). Notably, E-box motif sequences were not enriched on the Z chromosome compared to autosomes (Supplementary table 5) potentially, hinting to a sex-specific role in Z-gene regulation which could be explored in future studies.”

3. Line 130 says “The intersex line”... suggestion: change to “intersex individual” or “intersex cell line”.

We thank the reviewer for their comment. Following their recommendation, we have updated the naming of the intersex sample throughout the manuscript.

4. Discussion line 237-244. Here it is first concluded that Z-upregulation is driven by increased burst frequency, but not size, resembling the kinetic mode of the evolutionary distinct X-upregulation observed in mouse. Then it is stated that the core promoter elements controlling transcriptional burst size remain the same. What is then the relevance of the latter molecular mechanism in this context?

To clarify, the primary focus of our study is avian dosage compensation. However, since we present the first-ever allele-resolved characterization of transcriptional burst kinetics in birds, we also took the opportunity to explore fundamental aspects of burst kinetics, such as the genomic regulation of burst size and frequency. Although the genomic encoding of transcription burst size is not directly relevant in the context of Z-chromosome dosage compensation, we found it worthwhile to report that the fundamental principles regulating burst size remain the same between birds and mammals, despite the 310 million years of evolution separating the two. We found this result particularly interesting, as it points to evolutionary conservation of kinetic modalities. This result may therefore provide valuable information to the research community in transcriptional burst kinetics.

5. Discussion lines 256-257 conclude that Z-upregulation results from female cells carrying only one Z allele, not the presence of W. Lines 263-265 says “Intriguingly, similarly Z-enriched ribosomal density was observed in ZZW intersex cells suggesting the involvement of W-linked factors”. Comment: I agree that the apparently nuanced role of the W chromosome (i.e. not promoting Z-upregulation but possibly promoting increased translational efficiency of Z-transcripts in females through higher ribosomal density) is

interesting. Is there any idea how this mechanism could be associated with the W chromosome?

We agree with the Reviewer that this is a very interesting point that warrants more study. Although extensive investigation of how the presence of a W chromosome could affect the translation efficiency of Z-linked transcripts is beyond the scope of our work, we speculate that one or multiple W-linked factors may be implicated in translational regulation. For example, the chicken W encodes for long non-coding RNAs and miRNAs most of which have not been extensively characterised. In this instance, lncRNAs may act as sponges (Statello et al. 2021, *Nature Reviews Molecular Cell Biology*), sequestering miRNAs that target Z-linked transcripts (such as miR-2954, expressed in both males and females) to prevent translation inhibition caused by miRNA-mRNA interactions. Even more intriguingly, miRNAs have also been reported to enhance translation or even induce translational activation in some instances (O'Brien et al. 2018, *Frontiers in Endocrinology*, Ørom et al. 2008, *Molecular Cell*, Zhang et al. 2014 *Cell*).

We have included a passage in the Discussion section to outline some of our hypotheses. The passage reads as follows (Lines 326-329):

“This could include mechanisms involving lncRNAs acting as sponges to sequester miRNAs⁵⁶ targeting Z-linked transcripts, thereby preventing mRNA-miRNA interactions that result in translation inhibition, or even miRNAs that enhance translation of Z-linked transcripts depending on their localization and 5' UTR structures⁵⁷⁻⁵⁹.”

6. Discussion lines 272-275, conclude that there is an unexpected similarity to mammalian dosage compensation when all regulatory layers are coherently taken into consideration.

Comment: In what way is Z chromosome dosage compensation unexpectedly similar to X chromosome dosage compensation? In terms of overall compensation or in terms of the interplay between the underlying mechanisms?

Both. While chicken lack the extra layer of dosage compensation that X-inactivation provides in mammals, the features as well as magnitude of dosage compensation Z-upregulation provides for a single allele is highly similar to what is observed for X-upregulation. From a mechanistic point of view, the increased burst frequency, combined with the increased translation efficiency of the single female Z are remarkably similar to X-upregulation of the single active X in males (XaY) and females (XaXi) and increased translation efficiency of X-linked transcripts in both male and female mammals. We have added the following text to more clearly state this:

“Notably, however, while X-linked RNA levels are almost fully balanced between mammalian female and males, the transcriptional upregulation of their single active X (X_aX_i and X_aY , respectively) does not fully reach the biallelic RNA levels of autosomes^{29,30}. Consequently, the enhanced Z translation rate alongside transcriptional Z upregulation in female birds (ZW) parallels the scenario in mammalian cells, across the sexes. In contrast male birds (ZZ) exhibit balanced Z-to-autosome RNA levels, deviating from this pattern. Thus, our data highlight fundamental principles of dosage compensation that apply across evolutionarily distinct sex-chromosome systems.”

7. Abstract. Clarify which of the revealed mechanisms that are “remarkable conserved” and which mechanisms that are “new”. In what way is this study highlighting the importance of gene-dosage balance across diverse species?

We thank the reviewer for their suggestions and have updated the abstract to better convey this. Sex dosage is crucial for embryonic viability in mammals (Cecalev et al. 2024, *Development*, Borensztein et al. 2017 *Nature Structural and Molecular Biology*, Marahrens et al. 1997, *Genes and Development*), yet birds have for a long time been thought to largely lack dosage compensation (Baverstock et al. 1982, *Nature*, Ellegren et al. 2007, *BMC Biology*, Mank 2013, *Trends in Genetics*). Our study demonstrates that birds, do in fact implement sex chromosome dosage compensating mechanisms at the transcriptional and translational levels, achieving a high degree of Z-chromosome compensation at the proteomic level suggesting that sex chromosome dosage compensation in birds may be as crucial as it is for mammals.

We have updated the abstract section of our manuscript to better convey our message. The passage now reads:

“Remarkably, our data reveal that females upregulate their single Z chromosome through increased transcriptional burst frequency, mirroring mammalian X upregulation. Z-protein levels are further balanced in females through enhanced translation efficiency. Additionally, we present the first global analysis of promoter elements regulating transcriptional burst kinetics in birds, revealing a striking conservation of the genomic encoding of burst kinetics between birds and mammals.

Our study provides unique insights into the regulation of avian dosage compensation, and when considering all regulatory layers collectively, an unexpected similarity between avian and mammalian dosage compensation becomes apparent.”

We would like to express our gratitude once again to Reviewer 2 for their insightful comments, which we believe have been thoroughly addressed. We hope the revised version of our paper is now deemed suitable for publication.

RESPONSE TO REVIEWERS' COMMENTS

Reviewer #1 (Remarks to the Author):

This is a revised version of a work characterizing the mechanism of dosage compensation, if any, of birds, using multi-omics approaches on chicken cell lines. I don't think the authors have fully addressed my previous questions and I detailed my comments below. The limitation of novelty and the necessity and value of using a triploid cell line are my previous major concerns. Instead of answering directly to the questions, the authors reiterated how much new data they have generated. This does not justify the novelty but reflects lack of clear hypothesis and corresponding experimental design. Again I acknowledge the discovery of transcriptional bursting in chicken, in parallel to that of mammals, which I think is important for such a convergent pattern. However, without knowing the clear targeting mechanism on the X in mammals, or on the Z in chicken, based on what the authors have answered in this revision, it is rather descriptive. In this revision, the authors also found a contradiction compared their results to Lester et al. PNAS, which confounds the conclusion of this work. Finally I think the title "...in birds" is misleading and an overstatement, given that only cell lines of chicken were studied here.

We thank the reviewer for the additional comments and appreciate the opportunity to clarify points that may have caused confusion. While we address each specific concern point-by-point in the following sections, we would first like to respond to some of the broader remarks. We found aspects of the review somewhat unconventional and, at times, lacking in precision—particularly in the referencing of our current work, prior literature, and the present state-of-the-art of the field and technical methodology. Additionally, we found the review overly opinion-based while lacking in technical questions to address. That said, we have devoted considerable time and effort to thoroughly and thoughtfully respond to every comment, and we sincerely hope that the revised manuscript will now be considered suitable for publication.

Specific points:

(1) We strongly disagree with the assertion that our study “*lack of clear hypothesis and corresponding experimental design*”. On the contrary, the rationale and objectives of this study are well-defined: to investigate the extent and mode of sex-chromosome dosage compensation in birds. To this end, we applied a comprehensive multi-omics approach importantly at allelic resolution, in contrast to previous studies. Our experimental strategy is suited to investigate gene dosage at both the level of chromatin, transcriptional bursting kinetics, allelic RNA levels, translational rates, and finally protein levels. We emphasize that the use of an allele-resolved system is absolutely essential in properly dissecting dosage compensation, something that has been appreciated for decades in the field X-chromosome dosage compensation (e.g. allele-specific X-inactivation and X-active-specific upregulation) but has been lacking in studies in birds. Why? Because multiple *different* allelic expression modes can lead to the seemingly identical non-allelic “total” gene expression level (e.g. one allelic going up, and the other one down as in mammalian X-inactivation/X-upregulation; both alleles being dampened in expression as in X-chromosome dampening in the early human embryogenesis etc.). Indeed, this allelic resolution was critical to identifying female-specific Z-upregulation and its kinetic regulation via increased transcriptional

burst frequency, a mechanism strikingly convergent with X-chromosome regulation in mammals. We firmly maintain that our study offers both significant biological insights and a valuable data resource, supported by a well-chosen experimental approach.

(2) We appreciate the reviewer's acknowledgment of the significance of discovering transcriptional bursting in chicken. However, we would also like to emphasize that this is only one of several novel findings in our study, most importantly female Z-linked upregulation at the RNA and translational levels.

(3) The reviewer states that "*the authors also found a contradiction compared their results to Lester et al. PNAS, which confounds the conclusion of this work*". – We don't understand why obtaining different results from a previous study (Lester et al. PNAS 2024) would make our work "confounded". We have already addressed the Lester study in depth in our previous response letter (indicating lower power in Lester et al) as well as in additional figures in the manuscript, and we will return to this question again in the current letter.

(4) Finally, we acknowledge the reviewer's concern about the title. However, the statement that our study was limited to chicken cell lines is factually incorrect. In addition to cell lines, our analyses include six in vivo chicken tissues, all studied at allelic resolution (see main Figure 1). Nonetheless, we agree that "birds" as a general term may suggest comparative work across avian species, which was not performed here. Although the implications of our study clearly extend beyond chickens specifically, we have revised the title in accordance with the reviewer's suggestion. Ultimately, we defer to the publisher's discretion regarding the final title.

New title:

"Multi-layered Z-chromosome upregulation by increased transcriptional burst frequency and elevated translational rates in chicken"

1. About the novelty. A general issue is that the authors tried to emphasize what they have done, instead of why this is necessary or important in their response of justifying the novelty. If no clear hypothesis or aim is laid out for massive work, it reflects a poor design of experiment.

1) Indeed, the authors generated bulk RNA-seq, scRNA-seq, ATAC-seq, ChIP-seq, Ribo-seq and MS data in this work. But these techniques are as regular as PCR nowadays for any molecular genomics labs, and one cannot appreciate the difficulty of collecting these data from chicken.

The sweeping comment regarding "poor design of experiment" is, in our view, inappropriate. We have clearly outlined why an allele-specific design is not only suitable but essential for studying sex chromosome dosage compensation. Furthermore, it is misleading of the reviewer to equate the generation of bulk RNA-seq, scRNA-seq, ATAC-seq, ChIP-seq, Ribo-seq, and mass spectrometry data with routine PCR. Although we agree w that several of these techniques are nowadays widely available (except probably Smartseq3Xpress [providing full-transcript-length read coverage with allelic resolution in single cells] and Ribo-seq [providing translation rates]), we respectfully disagree that utilizing all these methods, spanning the omics layers, for current studies renders them worthless. We have outlined the reasons for using these methods throughout the text. Specifically, in this work, we characterise the presence and extent of Z chromosome dosage compensation, across multiple

regulatory layers in a way and to an extent that has not been implemented before. Please note, once again, that the sequencing-derived data in our paper has allelic resolution – which is necessary for addressing the core question, which not available before, and is not trivial to generate, requiring the right breeding stock and facilities: To attain high resolution of SNPs for allelic analyses, we crossed the wild species Red junglefowl [*Gallus gallus*] with domesticated chicken White Leghorn [*Gallus gallus domesticus*].

That *others could in principle* generate similar datasets does not diminish the significance of our work—we *have done so*, and in doing this, have provided the first comprehensive, multi-layered, allele-resolved analysis of Z chromosome dosage compensation in birds.

2) Why is it important to study pure-bred and cross-bred chicken, if the question of the project is to study whether there exists and what is the mechanism of dosage compensation in birds? What is the explanation if there is, or not difference between pure-bred vs. cross-bred chicken in the findings of dosage compensation?

The point of using two strains of chicken is to achieve allelic resolution. Although inclusion of the pure maternal/paternal lines is not a main objective in our study, it is commonplace and suitable to include pure breeds in allelic analyses e.g. for confirmation of SNVs assessment of allelic accuracy (as we have done).

We take this question as yet another opportunity to highlight the importance of allelic resolution in studying dosage compensation. To our knowledge, no prior study has investigated sex chromosome dosage compensation using allelic resolution in chicken, or birds in general. In previous work, we have elaborated on the importance of allelic resolution in characterizing the transcriptional dynamics of X-chromosome upregulation in mouse, demonstrating that lack of allelic resolution fails to capture major regulatory mechanisms and dynamics of X-chromosome dosage compensation during early embryonic development in mouse (Lentini and Reinius, 2023 *Curr Bio*, Lentini et al. 2022 *Nat Comms*, Larsson et al. 2019 *Nat Struct Mol Bio*). Additionally, correct interpretation of dosage compensation effects requires knowledge of underlying sex chromosome copy numbers as well as reliable quantification, as seen in *Drosophila* (Straub et al. 2005). Finally, allelic resolution is crucial for the inference of transcriptional kinetic modalities (Larsson et al. 2019 *Nature*), namely burst frequency and burst size, the investigation of which constitutes an important part of this current work. In this study, we have used both pure-bred and cross-bred chicken to examine Z-chromosome dosage compensation. While the use of pure-bred chicken lacks the allelic resolution necessary to dissect the regulatory dynamics of the two Z chromosomes in males, it is sufficient to show Z-chromosome upregulation of the female Z, due to the presence of one single copy. However, the allelic resolution provided by the use of cross-bred chicken is crucial to understanding whether the two male Z chromosomes are differentially regulated, to directly and accurately compare the transcriptional output of the single female Z chromosome to each of the two male Z chromosomes, to examine the accessibility of each of the male Z chromosomes and

crucially, to dissect the transcriptional kinetic modalities of the Z chromosome. The importance of allelic resolution is emphasized throughout the manuscript text:

Lines 101 – 107:

“Interestingly, utilizing our allelic expression measurements, we observed that the single Z chromosome in female tissues was distinctly upregulated compared to individual autosomal alleles as well as compared to the separate transcriptional output of each male Z allele (Fig. 1e, Supplementary Fig. 1f), indicating that partial dosage compensation is achieved through hyperactivation of the female Z chromosome.”

Lines 167-172:

“To explore Z-upregulation at cellular allelic regulation, we performed full-transcript-length scRNA-seq using Smart-seq3³⁸ deep-sequencing of CEF cell lines (Supplementary Fig. 7a). These data reconfirmed female-specific Z-chromosome upregulation, but here, importantly, at the level of individual allele in single cells (Supplementary Fig. 7b). Notably, biallelic expression of the MHM region was observed in intersex ZZW cells (Supplementary Fig. 6c), thus demonstrating its expression from the maternal and paternal alleles.”

Lines 188 – 195:

“At the single-cell level, eukaryotic transcription is inherently stochastic and occurs in short bursts of activity from the individual alleles^{41,44}. Transcriptional kinetics can be represented by burst frequency (rate of transcription pulses) and burst size (average number of molecules produced per burst) within the two-state telegraphic model of transcription. We and colleagues, previously characterized the genomic encoding of transcriptional burst kinetics in mouse from allelic scRNA-seq data¹⁹. Moreover, we showed that X-upregulation in mouse is primarily driven by increased transcriptional burst frequency, and not increased burst size^{30,31}.”

Lines 207-213:

“We next explored the kinetic modulus of Z-upregulation, observing that burst frequency was increased on Z relative to autosomes in female ZW cells carrying either Z allele ($P_{WL-allele} = 2.68 \times 10^{-4}$, $P_{RJF-allele} = 0.029$, MWU Test) whereas burst frequency was not increased in Z alleles in male ZZ and intersex ZZW cells (RJF alleles analysed in triploid cells) (Fig. 2i). Conversely, burst size remained close to autosomal levels for all cells ($P > 0.41$) (Supplementary Fig. 10a), indicating that transcriptional Z-upregulation is primarily driven by burst frequency. It should be noted that inference of transcriptional kinetics is only robust for individual alleles¹⁹, necessitating allelic scRNA-seq data herein provided, and allowing accurate kinetic inference of RJF alleles but not WL alleles in our triploid CEF cell line.”

3) This is related to the question that I raised in the last round of revision. There is no indication at all that Y chromosome, or W chromosome would impact the dosage compensation from any species. So it only reflects a poor design of experiment and lack of hypothesis, when the authors decided to study a ZZW cell line and examine whether there is allelic expression or not. The authors mentioned below ZZW is from a wild type chicken, but how rare is that? This is like saying to study an aneuploidy cell line so that one can understand the mechanism of gene regulation in a normal individual without aneuploidy, which does not make sense.

Triploid line:

The reviewer asked this in the first round of revision, to which we have responded extensively. Triploid chickens, both ZZZ and ZZW, are rare but naturally occurring (~0.1 -

0.5% of cross-bred chickens), as we have outlined in the manuscript text since the beginning, lines 115-117:

“Triploidy is a naturally viable, but exceedingly rare, genotype in chickens (0.1–0.5%) arising from chromosomal nondisjunction during oogenesis³² (Supplementary Fig. 2f), and, curiously, was first described by none other than Susumu Ohno³³.”

Importantly, these chickens are **not** aneuploid, as states by the reviewer. They are **chromosomally balanced** triploid chickens (either 3N- ZZZ or 3N-ZZW like in this current work). 3N-ZZW chickens phenotypically resemble females until approximately week 20 post-hatching as shown in Lin et al. 1995.

Effects of sex-chromosome complement:

The statement *“There is no indication at all that Y chromosome, or W chromosome would impact the dosage compensation from any species”* is incorrect. Leaving the mammalian Y-chromosome aside and focusing on the avian ZW system, there is indeed a clear hypothesis in the field that the W chromosome may be contributing to Z-chromosome upregulation. This theory was put forth by Jennifer Marshall Graves in her 2003 work (Graves 2003, *Cytogenetic and Genome Research*), not to mention the work by Teranishi and colleagues, suggesting a role of the W chromosome in the methylation patterns of the MHM region of the Z chromosome in triploid chickens (Teranishi et al. 2001). Teranishi and colleagues conclusively show that the presence of the W chromosome is correlated with hypomethylation of the MHM region in triploid chicken, which our work confirms. How then can we exclude the possibility that the presence of a W chromosome is irrelevant to other layers of regulation of the Z chromosome, such as Z-chromosome upregulation?

We therefore used this ZZW sample, which harbors both a W chromosome, like a typical ZW female, as well as two Z chromosomes, like a typical ZZ male to dissect Z-chromosome upregulation from either of the Z chromosomes by utilizing the allelic resolution that a cross-bred chicken breed provided us with, suggesting that unlike the hypothesis put forth by the aforementioned work (Graves 2003) and evidence for the W chromosome controlling the MHM region (Teranishi et al. 2001), neither of the Z chromosomes in the ZZW sample, show upregulation.

We have updated the manuscript text to read as follows, lines 117-124:

“It has been previously suggested that the presence of a W chromosome may contribute to the control of Z chromosome upregulation³⁴, as ZZW individuals have been previously shown to express the female-specific Z-linked MHM region¹³. The presence of triploid cells thus enabled us to assess the extent of Z-linked dosage compensation across diploid males (ZZ:AA), diploid females (ZW:AA), and triploid intersex individuals (ZZW:AAA) as well as to uncouple a) the effect of the W chromosome on Z-upregulation, in the presence of two copies of the Z chromosome and b) the effect of biological sex on Z-upregulation.”

And lines 126-129:

“Interestingly, whereas RNA-seq of CEF cell lines reconfirmed Z-upregulation in ZW females on par with in vivo tissues (mean fold-change 1.54 in CEFs, Fig. 2b), no such upregulation was observed in ZZW, but instead, a marked buffering of autosomal gene expression (Supplementary Fig. 3b), i.e. less than linearly additive, as has been observed for other species³⁵, suggesting that Z-chromosome upregulation is not dependent on the presence of a W chromosome, as has been previously hypothesized³⁴.”

4) This is a novel point, as I mentioned in the previous comment and also here above.

OK.

5) The same as 4)

OK.

6) Again the authors were describing what they did without elucidating why they did it. In addition, from ATAC-seq and ChIP-seq, the authors found no difference between male and female Z chromosome, which provides no evidence for dosage compensation from these two datasets at all! How generating a new data without generating a new and relevant conclusion can justify the novelty of the work?

This comment appears to reflect a misunderstanding of how transcriptional regulation is interpreted from chromatin data. It is well established that chromatin accessibility and histone modifications do not always correlate directly with gene expression levels, especially when examining broad chromosomal patterns. The absence of global male–female differences in ATAC-seq and ChIP-seq signals across the Z chromosome does not imply the absence of dosage compensation; rather, it highlights that such compensation may operate through mechanisms beyond those specific regulatory layers. Chromatin accessibility, as measured by ATAC-seq, typically captures major differences in regulatory element usage and lineage-specific developmental states and lineage choices. While we did not observe global Z-chromosome accessibility differences between sexes, we did detect clear regional differences, such as the female-specific accessibility of the MHM locus—underscoring that the data are informative. Regarding ChIP-seq, we analyzed key active histone marks (H3K4me3, H3K9ac, H3K27ac, H4K16ac). The lack of sex differences in these marks over the Z chromosome does not invalidate the study. On the contrary, it suggests that transcriptional upregulation in females occurs in the absence of chromatin-level enrichment for these particular modifications. While we would have welcomed the opportunity to profile a broader set of chromatin marks, practical limitations necessitated prioritization during data collection. Nevertheless, the integration of these data with allele-resolved scRNA-seq, Ribo-seq, and proteomics provides a comprehensive and multi-layered view of Z dosage regulation not previously available.

We indeed already included discussions about the interpretation of the chromatin data in our manuscript text in the **Discussion** section and have further expanded in this revision round (with new additions shown in red):

Lines 297-302:

“Chromatin accessibility and permissive histone modifications have been suggested to at least partially underlie X-upregulation in mouse^{52–54}. However, despite a pronounced compensation on the transcriptional level, using allele-resolved ATAC-seq and quantitative ChIP-seq, we observed no differences in accessibility or chromatin state between female and male Z-chromosomes, in line with previous reports of non-linear relationships between chromatin states and gene expression in X-upregulation^{31,52,53}, hinting to an additional level of similarity between mammalian and avian dosage compensation mechanisms.”

It has been previously suggested, that mammalian X chromosome upregulation may be partly driven by increased accessibility of the X chromosome (Talon et al. 2021). In previous work, we and others, observed no increased accessibility or significant changes in the chromatin landscape of the upregulated mammalian X chromosome (Yildirim et al. 2012, Lentini et al. 2020). However, chromatin accessibility and chromatin landscape (in the form of histone modifications) had not been previously examined in the context of sex chromosome dosage compensation in chicken. We therefore performed ATAC-seq and ChIP-seq to examine whether changes in chromatin accessibility and histone modifications accompany the transcriptional upregulation of the female Z-chromosome. We did not find any evidence to support that increased accessibility (ATAC-seq data) drives Z-chromosome upregulation. Instead, we observed significant differences at the transcriptional, translational and proteomic levels between the female and male Z chromosomes. Our observations that the female and male Z chromosomes do not significantly differ in chromatin accessibility and histone modification enrichment do not negate our work; instead, similar to our, and others', previous observations in mouse, this suggests a further point of similarity between mouse X chromosome upregulation and chicken Z chromosome upregulation. While we have previously outlined this in the previous version of our manuscript, we have made additional changes to clarify our aims:

The **Results** section used to read (lines 136-141):

“To gain further insights into the regulation of Z-upregulation we mapped chromatin accessibility in CEF lines using ATAC-seq. Similar to our previous findings of X-upregulation in mice³⁰, chromatin accessibility was not increased on the upregulated Z allele despite pronounced Z-upregulation on the transcriptional level (Fig. 2c and Supplementary Fig. 4a-c), suggesting limited involvement of chromatin accessibility in regulating Z-upregulation”

The **Results** section now reads (lines 137-139):

“To investigate whether the chromatin landscape of the Z chromosome contributes to the regulation of Z-upregulation we began by assessing chromatin accessibility in CEF lines using ATAC-seq. Similar to our previous findings of X-upregulation in mice³¹, chromatin accessibility was not increased on the upregulated Z allele despite pronounced Z-upregulation on the transcriptional level (Fig. 2c and Supplementary Fig. 4a-c), suggesting limited involvement of chromatin accessibility in regulating Z-upregulation.”

The **Discussion** section now reads (new additions in red, Lines 297-302):

*“Chromatin accessibility and permissive histone modifications have been suggested to at least partially underlie X-upregulation in mouse⁵²⁻⁵⁴. However, despite a pronounced compensation on the transcriptional level, using allele-resolved ATAC-seq and quantitative ChIP-seq, we observed no differences in accessibility or chromatin state between female and male Z-chromosomes, in line with previous reports of non-linear relationships between chromatin states and gene expression in X-upregulation^{31,52,53}, **hinting to an additional level of similarity between mammalian and avian dosage compensation mechanisms.**”*

7) So there is a contradiction between the Lester et al. vs. this work in their

conclusion. Basically Lester et al. from in vivo situation found evidence for dosage compensation, but the authors here did not, as the MS data here showed there is a sex-bias in protein level. (see more of my comments below)

We have addressed this point extensively during the previous round of revision and further expand on our explanations below. But briefly, both our study and Lister et al. 2024 found evidence for extensive dosage compensation both at the RNA and the protein levels. The main difference between our findings and those reported by Lister et al. 2024 refers to the extent of protein-level compensation provided by male-to-female expression ratios. While Lister et al. 2024 suggests a complete proteomic rebalancing (Male-to-Female protein ratio ~ 1), our data suggest that a small but significant male bias remains on the protein level (Male-to-Female protein ratio ~ 1.2). As described in depth in the previous point-by-point letter it is possible that this difference stems from a lack of power in Lister et al.

8) I don't think Ribo-seq on chicken is 'pioneering' at all: this was done in chicken five years ago (Zhong-Yi et al 2020 Nature)

It is true that Ribo-seq has been applied in chicken tissues before (Zhong-Yi et al. 2020). Importantly, however, Zhong-Yi et al. 2020 specifically used Ribo-seq to examine the evolution of transcriptomes and translomes across vertebrates, and Ribo-seq has *not been used to study Z chromosome dosage compensation*, to our knowledge.

9) See point 3) above.

2. I asked in the last round regarding the targeting mechanism of Z that leads to transcription burst preferentially occurred in female but not male chicken. The authors mentioned E-box transcription factor as a potential responsible mechanism. However, during this revision, the authors indicated that this transcription factor is not female-biased from transcriptomic data. The binding motifs of E-box TF are also not enriched on the Z vs. autosomes. Then how would this explain the role of E-box transcription factor, if any, in targeting specifically the female Z chromosome? The only suggestive evidence seems to be 'a preferential enrichment of E-box-binding transcription factor' on the female Z relative to male Z. But I did not find in the method (it only showed the motif search) how this conclusion was reached, did the author perform CHIP targeting the E-box-binding TF in male and female cell lines?

While it is true that we did not observe a difference in the number of motifs for the Z chromosome compared to autosomes, we did observe an enrichment of E-box motif-binding factors on the female and intersex Z chromosomes. As no Z-chromosome upregulation was observed for the triploid ZZW sample, E-box TF enrichment is unlikely to be a driver of the increase in Z-chromosome transcriptional burst frequency observed in females. However, we do observe a transcriptional similarity between female and intersex samples, potentially suggesting that the enrichment of bound E-

box TFs may be involved in sex-specific gene regulation. This has been updated and outlined in the manuscript text in the previous round of revision:

“Notably, E-box motif sequences were not enriched on the Z chromosome compared to autosomes (Supplementary Data 5) potentially, hinting to a sex-specific role in Z-gene regulation which could be explored in future studies”

To provide further clarification to this point, we have further updated the respective text to read as follows, lines 150-154:

“Notably, E-box motif sequences were not enriched on the Z chromosome compared to autosomes (Supplementary Data 5) potentially, hinting to a sex-specific role in Z-gene regulation, rather than the female-specific Z-upregulation we observed, something that could be further explored in future studies.”

Finally, as we interpret the observed enrichment as potentially a sex-specific gene regulation effect, and not directly related to the mechanisms of Z-upregulation, the underlying mechanisms targeting E-box TFs to the Z chromosome in females and intersex chicken is beyond the scope of this work.

Regarding the preferential enrichment of E-box binding transcription factors: Our observation of E-box TF enrichment was based on ATAC-seq transcription factor footprinting and not ChIP-seq, as we have clarified from the beginning, lines 142-145.

“Through a global analysis of transcription factor footprinting using ATAC-seq data, we examined differential transcription factor binding by performing pairwise comparisons of the Z chromosome and autosomes between the sexes.”

Details of how this analysis was performed are described in the Methods section (lines 727-735) under **ATAC-seq: Transcription factor footprinting and identification of differentially bound transcription factors.**

3. The authors answered my second question about the targeting mechanism to the X by citing their published work, except for how exactly the X chromosome is targeted for transcription bursting. I wonder what transcription factors (supposed to be male biased or specific) and binding motifs are involved in mammalian X transcription bursting?

The exact mechanisms behind targeted transcriptional bursting of the X chromosome are still unknown and an active field of research. Although we agree that understanding which factors contribute to increased X-specific burst frequency would be intriguing, it is beyond the scope of this work focusing on Z-upregulation.

4. A key difference between this work vs. Lister et al. 2024 is that based on MS data, the authors find in this work a male biased translation pattern, while Lister et al. find a nonbiased pattern. This makes the results of this work DO NOT support dosage compensation in the translation level, while Lister et al. does. The authors attributed the difference to more sensitive MS results from this work, but then the conclusion is confusing. Does this mean dosage compensation in translation only occurs for highly translated protein? In addition, the Lister et al. 2024 used in vivo

tissues, while the authors in this work did MS on cell lines. This is also an important confounding factors, and I generally tend to think an in vivo tissue likely reflect better the dosage compensation than the cell lines.

In their work, Lister et al.2024 suggest that the incomplete dosage compensation between the male and female Z chromosomes observed on the transcriptomic level, is completely resolved at the proteomic level. Our work suggests that the incomplete dosage compensation observed on the transcriptional level is resolved **to a significant, although not complete**, degree at the proteomic level. **Crucially, we would like to highlight that both Lister et al. 2024 and this current study suggest a significant further rebalancing of dosage on the protein level.** These findings, in addition to our findings of increased translational efficiency, suggest extensive dosage compensation at the translation level.

We have thoroughly re-analysed the data provided by Lister et al. 2024 and provided a direct comparison to our findings. Lister et al.2024 conclude that, at the protein level, no significant difference is detected between male:female ratios of autosomes and the Z-chromosome. However, it is crucial to highlight that the low degree of correlation between RNA and protein measurements, in addition to the relatively low number of proteins detected in Lister et al.2024, may explain the conclusion of non-significant difference between autosomes and the Z chromosome in females.

We summarize our response in lines 319 – 327:

“Our proteomic experiment was similar to that of a recent study reporting the complete lack of sex differences in Z-protein levels in chicken and platypus⁵⁵. However, in contrast, our mass spectrometry data revealed a small but significant male bias remaining at the protein level (male-to-female expression ratio: 1.2-fold, $P < 8.6 \times 10^{-7}$). This discrepancy in results may stem from differences in tissues analysed (liver and heart versus fibroblasts) or the increased statistical power of our experiments, capturing more than twice the number of proteins and maintaining a strong correlation between RNA and protein ratios ($\rho = 0.750505$, $P < 2.2 \times 10^{-16}$, MS3 data) (Supplementary Fig. 11b and Supplementary Fig.14a-b).”

Additionally, there is no evidence to suggest that more highly translated proteins are subjected to better dosage compensation as there is no correlation between high protein abundance and male:female protein expression ratios for either the autosomes or the Z chromosome.

The reviewer further suggests that “I generally tend to think an in vivo tissue likely reflect better the dosage compensation than the cell lines.” without providing accompanying references. In this work, we have analysed both in vivo and in vitro data to show that the degree of transcriptional dosage compensation is highly comparable between the two, lines 126-127:

“Interestingly, whereas RNA-seq of CEF cell lines reconfirmed Z-upregulation in ZW females on par with in vivo tissues (mean fold-change 1.54 in CEFs, Fig. 2b)...”

5. For the answers to my previous question 4, the authors justified using a ZZW cell line because ‘intersex sample allowed us to evaluate whether the W chromosome contributes to the Z-linked dosage compensation’. But why there is a need to evaluate this? As I previously asked, there was no evidence or reasoning to think that the Y or W would impact dosage compensation given both chromosomes have few functional genes. Could the authors elaborate by what mechanisms the Y/W would impact dosage compensation? And why it is important to examine whether Y/W would have such an impact, if any. Also, the authors also responded “increased Z chromosome burst frequency in females comes from having only one Z copy”. I did not understand, what is the alternative given female only have one Z, and if there is a Z-linked burst, it must come from the only Z.

As mentioned in Question 1, point 3, this question was previously raised by the reviewer in the last round of revision, to which we have answered thoroughly and extensively. We expand here further. The ZZW individual was a fortunate coincidence during our sampling and provides unique allelic-level, multiome data. While not perfect, it offers a valuable opportunity to begin disentangling the effects of sex, ZZ dosage, and the presence of the W chromosome. We, however, realized that some sections of the text regarding the ZZW individual came out somewhat unclear - Please see our response to Reviewer 2’s remaining question.

The ZZW individual provides a unique opportunity to conclusively investigate whether the W chromosome has an effect on Z-chromosome upregulation. The current study is

not the first to suggest that this may be the case. In fact, an extensive essay by Jennifer Marshall Graves, (Graves 2003, *Cytogenetic and Genome Research*) goes into extensive detail into the hypothesis of a W-linked factor being implicated in the regulation of Z chromosome upregulation. **Excerpt from Graves 2003:**

“However, the presence in chickens of the MHM locus on the Z whose methylation depends on the presence of the W (Teranishi et al., 2001), suggests that the W chromosome may play a part in upregulating Z chromosome activity. Methylation of the MHM sequence is evidently under the control of a factor on the W chromosome, called F by the authors but here named ZUF for Z Upregulating Factor. If MHM is involved in dosage compensation, this W-borne factor could control upregulation of the single Z in ZW birds.”

Due to the rarity of ZZW chickens, especially cross-bred ZZW chicken, that would allow one to examine the transcriptional dynamics and regulation of the two Z chromosomes separately, through allelic resolution, the question of whether the W chromosome plays any part in Z chromosome upregulation remained opened. Therefore, by examining the presence of Z chromosome upregulation in this sample, we are able to uncouple the effect of having two Z chromosome copies from the presence of the W chromosome.

Regarding as to how the W chromosome could participate in the regulation of Z-dosage compensation: The discussion section of the manuscript text was expanded in the **last round of revision** in order to address that. Since the ZZW sample showed no transcriptional Z-upregulation, but did show increased Z-translation, we have provided hypotheses as to how the W may contribute to this, lines 328-335:

“What mechanisms drive additional dosage compensation at the proteome level? Our Riboseq data indicates that this is least partially achieved by higher ribosomal density, and thus increased translational efficiency, of Z-transcripts in females. Intriguingly, similarly Z-enriched ribosomal density was observed in ZZW intersex cells suggesting the involvement of W-linked factors. This could include mechanisms involving lncRNAs acting as sponges to sequester miRNAs⁵⁶ targeting Z-linked transcripts, thereby preventing mRNA-miRNA interactions that result in translation inhibition, or even miRNAs that enhance translation of Z-linked transcripts depending on their localization and 5' UTR structures⁵⁷⁻⁵⁹.”

Regarding “increased Z chromosome burst frequency in females comes from having only one Z copy”: this sentence aims to highlight that it is the presence of only one Z chromosome that leads to increased burst frequency and NOT the presence of the W.

We thank Reviewer #1 for their input, and we hope that the Reviewer now finds our study suitable for publication.

Reviewer #2 (Remarks to the Author):

I think that the authors have made an overall good job in revising their manuscript. The changes made have increased the clarity of the study. I only have one remaining concern.

1. This is about the interpretations of the expression patterns found in the triploid

intersex individual (ZZW:AAA). The reduced expression of the autosomal genes is interpreted as a mechanism for dosage compensation of the Z chromosome. However, a reduced expression of the autosomal genes could also be a mechanism for buffering autosomal expression per se to maintained function and viability of triploid individuals. In their answer the authors say “ We do agree that buffering of autosomal expression may be related to maintained function and viability of triploid individuals”. However, in the changed text in the manuscript the reduced expression of autosomal genes is still only viewed as a mechanism of Z-chromosome upregulation. Thus, some further clarifications of why this is the only or most likely interpretation is needed.

We thank the reviewer for their positive assessment of our revisions and for raising an important point regarding the interpretation of autosomal gene expression patterns in the triploid intersex individual (ZZW:AAA).

We fully agree that the observed reduction in autosomal gene expression could reflect a general buffering mechanism to maintain functional gene dosage and viability in triploid individuals (or other unknown mechanisms), rather than being interpreted as a form of Z-chromosome dosage compensation. It was not our intention to present the observed pattern as a mechanism for Z-chromosome dosage compensation. We appreciate the reviewer highlighting this ambiguity, and we have reworded the relevant sections of the manuscript to reflect a more balanced and cautious interpretation. The ZZW individual represents a rare and fortunate sampling event, and while several aspects of its biology remain unclear, we believe it offers unique and valuable data—both for understanding the consequences of triploidy and for disentangling the effects of sex, Z/ZZ chromosome dosage, and the presence of a W chromosome.

We believe the reviewer is referring to this section in the manuscript text:

“Interestingly, whereas RNA-seq of CEF cell lines reconfirmed Z-upregulation in ZW females on par with in vivo tissues (mean fold-change 1.54 in CEFs, Fig. 2b), dosage compensation in the ZZW intersex individual was not mediated through Z-upregulation but through buffering of autosomal expression (Supplementary Fig. 3b), i.e. less than linearly additive, as has been observed for other species³³, suggesting that Z-chromosome upregulation is not dependent on the presence of a W chromosome.”

We agree that this section could benefit from reformatting and additional clarity, as the sentence “dosage compensation in the ZZW intersex individual was not mediated through Z-upregulation but through buffering of autosomal expression” gives the impression that we are interpreting autosomal buffering as a mechanism of sex chromosome dosage compensation. We assure the reviewer that that was not our meaning, but rather, that the autosomal buffering results in improved stoichiometric balance.

We have updated the manuscript text, both in the **Results** and the **Discussion** sections, to better reflect our interpretations of this finding:

The **Results** section, used to read as follows (lines 124-129):

“Interestingly, whereas RNA-seq of CEF cell lines reconfirmed Z-upregulation in ZW females on par with in vivo tissues (mean fold-change 1.54 in CEFs, Fig. 2b), dosage compensation

in the ZZW intersex individual was not mediated through Z-upregulation but through buffering of autosomal expression (Supplementary Fig. 3b), i.e. less than linearly additive, as has been observed for other species³³, suggesting that Z-chromosome upregulation is not dependent on the presence of a W chromosome.”

It now reads (lines 126-131):

“Interestingly, whereas RNA-seq of CEF cell lines reconfirmed Z-upregulation in ZW females on par with in vivo tissues (mean fold-change 1.54 in CEFs, Fig. 2b), no such upregulation was observed in ZZW, but instead, a marked buffering of autosomal gene expression (Supplementary Fig. 3b), i.e. less than linearly additive, as has been observed for other species³⁵, suggesting that Z-chromosome upregulation is not dependent on the presence of a W chromosome, as has been previously hypothesized³⁴.”

The respective text in the **Discussion** section, used to read as follows:

“Conversely, we observed that incomplete dosage compensation in intersex cells was mediated by the reduced expression of autosomal genes, which would improve the stoichiometric balance and contribute to the viability of these individuals.”

This section now reads (lines: 307-310):

“Conversely, we observed that intersex cells displayed a buffering of autosomal gene expression, which would improve the overall stoichiometric balance, rendering Z-upregulation redundant, across both autosomal and sex chromosomes, and therefore contribute to the viability of these individuals.”

We thank Reviewer #2 for their constructive feedback and thoughtful review. We hope that the revised manuscript now meets the standards for publication. We firmly believe that this study represents an important milestone, both as a valuable multiomic resource and in advancing our understanding of dosage compensation in birds.

Point-by-point responses to the reviewers' comments

Reviewer #2 (Remarks to the Author):

I am happy with the authors' response to my last issue raised.

We thank the reviewer for finding our revision satisfactory.

Remaining main objections from reviewer 1 are that the authors lack clear hypothesis and highlight their methods rather than pinpointing what their novel findings are (i.e. that they describe what they have done rather than why). I think that the authors in general have managed to solve these issues in the revision of the ms. However, some parts of the Discussion section would benefit from further increased clarity.

Thank you for noting that Reviewer 1's questions are resolved. Following your suggestions, we have now made the suggested clarifications in the Discussion, as detailed below.

1.Lines 296-302 in the discussion. This paragraph starts with “Chromatin accessibility have been expected to at least partly underlay X-upregulation in mouse....” the authors then explains what they find: ...” we observed no differences in accessibility or chromatin state between female and male z-chromosomes” and conclude that this hints to an “additional level of similarity between mammalian and avian dosage compensation mechanisms.”

Suggestion: This paragraph would benefit from clarification. Is the main similarity between birds and mammals that chromatin accessibility is not a major mechanism of X/Z-upregulation? (even if some early studies suggested so in mouse).

Thank you for highlighting this point. We have now expanded this section to clarify our message. It now reads:

“Chromatin accessibility and permissive histone modifications have been suggested to at least partially underlie X-upregulation in mouse^{52–54}. However, despite a pronounced compensation on the transcriptional level, using allele-resolved ATAC-seq and quantitative ChIP-seq of selected markers, we observed no differences in accessibility or chromatin state between female and male Z-chromosomes. This is in

line with previous reports of non-linear relationships between chromatin state and gene expression in sex-chromosome upregulation^{31,52,53}. The absence of chromatin accessibility differences between an upregulated and non-upregulated allele of both the mammalian X and avian Z chromosomes highlight another layer of similarity in their dosage compensation mechanisms.”

2. Lines 303-314 in the discussion nicely describes how the ZZW chicken was used (and why) but lack references to previous studies making it difficult for the reader to judge how novel the finding is. Even if this indeed is explained in other parts of the ms, it needs to be re-stated here to make the discussion stand by itself.

We thank the reviewer for this suggestion. While the relevant references were included elsewhere in the manuscript, we agree that restating them here improves clarity and allows the Discussion to be read independently. The text has been updated accordingly and now reads as follows:

“Conversely, we observed that intersex cells displayed a buffering of autosomal gene expression, which would improve the overall stoichiometric balance, rendering Z-upregulation redundant, across both autosomal and sex chromosomes, and therefore contribute to the viability of these individuals. Thus, we conclude that Z-upregulation by increased transcriptional burst frequency in female cells results from carrying only one Z allele, and not from the presence of a W chromosome, contrasting to the proposition of a locus on the W chromosome being the main driver of Z-upregulation³⁴.”

3. Minor suggestion 137-139: To investigate whether the chromatin landscape of the Z chromosome contributes to the regulation of Z-upregulation...”

Suggestion: Change to “To investigate whether the chromatin landscape of the Z chromosome contributes to Z-upregulation...”

We thank the reviewer for this helpful suggestion and have updated the text accordingly in the indicated passage:

“To investigate whether the chromatin landscape of the Z chromosome contributes to Z-upregulation, we began by assessing chromatin accessibility in CEF lines using ATAC-seq.”

We sincerely thank the reviewer for the final points of improvement and for all prior constructive feedback, which we believe have substantially strengthened the manuscript. We now look forward to its publication.